# DO NOT ESCAPE FROM THE MANIFOLD: DISCOVERING THE LOCAL COORDINATES ON THE LATENT SPACE OF GANS

**Jaewoong Choi**[*], **Junho Lee**[*], **Changyeon Yoon**, **Jung Ho Park**,
**Geonho Hwang**, **Myungjoo Kang**
Seoul National University
{chjw1475,joon2003,shinypond,jhpark009,hgh2134,mkang}@snu.ac.kr

## ABSTRACT

The discovery of the disentanglement properties of the latent space in GANs motivated a lot of research to find the semantically meaningful directions on it. In this paper, we suggest that the disentanglement property is closely related to the geometry of the latent space. In this regard, we propose an unsupervised method for finding the semantic-factorizing directions on the intermediate latent space of GANs based on the local geometry. Intuitively, our proposed method, called *Local Basis*, finds the principal variation of the latent space in the neighborhood of the base latent variable. Experimental results show that the local principal variation corresponds to the semantic factorization and traversing along it provides strong robustness to image traversal. Moreover, we suggest an explanation for the limited success in finding the global traversal directions in the latent space, especially $\mathcal{W}$-space of StyleGAN2. We show that $\mathcal{W}$-space is warped globally by comparing the local geometry, discovered from Local Basis, through the metric on Grassmannian Manifold. The global warpage implies that the latent space is not well-aligned globally and therefore the global traversal directions are bound to show limited success on it.

## 1 INTRODUCTION

Generative Adversarial Networks (GANs, Goodfellow et al. (2014)), such as ProGAN (Karras et al., 2018), BigGAN (Brock et al., 2018), and StyleGANs (Karras et al., 2019; 2020b;a), have shown tremendous performance in generating high-resolution photo-realistic images that are often indistinguishable from natural images. However, despite several recent efforts (Goetschalckx et al., 2019; Jahanian et al., 2019; Plumerault et al., 2020; Shen et al., 2020) to investigate the disentanglement properties (Bengio et al., 2013) of the latent space in GANs, it is still challenging to find meaningful traversal directions in the latent space corresponding to the semantic variation of an image.

The previous approaches to find the semantic-factorizing directions are categorized into *local* and *global* methods. The *local* methods (e.g. Ramesh et al. (2018), Latent Mapper in StyleCLIP (Patashnik et al., 2021), and attribute-conditioned normalizing flow in StyleFlow (Abdal et al., 2021)) suggest a sample-wise traversal direction. By contrast, the *global* methods, such as GANSpace (Härkönen et al., 2020) and SeFa (Shen & Zhou, 2021), propose a global direction for the particular semantics (e.g. glasses, age, and gender) that works on the entire latent space. Throughout this paper, we refer to these global methods as the *global basis*. These global methods showed promising results. However, these methods are successful on the limited area, and the image quality is sensitive to the perturbation intensity. In fact, if a latent space does not satisfy the global disentanglement property itself, all global methods are bound to show a limited performance on it. Nevertheless, to the best of our knowledge, the global disentanglement property of a latent space has not been investigated except for the empirical observation of generated samples. In this regard, we need a local method that describes the local disentanglement property and an evaluation scheme for the global disentanglement property from the collected local information.

---

[*]Equal contribution

In this paper, we suggest that the semantic property of the latent space in GANs (i.e. disentanglement of semantics and image collapse) is closely related to its geometry, because of the sample-wise optimization nature of GANs. In this respect, we propose an unsupervised method to find a traversal direction based on the local structure of the intermediate latent space $\mathcal{W}$, called *Local Basis* (Fig 1a). We approximate $\mathcal{W}$ with its submanifold representing its local principal variation, discovered in terms of the tangent space $T_{\mathbf{w}}\mathcal{W}$. Local Basis is defined as an ordered basis of $T_{\mathbf{w}}\mathcal{W}$ corresponding to the approximating submanifold. Moreover, we show that Local Basis is obtained from the simple closed-form algorithm, that is the singular vectors of the Jacobian matrix of the subnetwork. The geometric interpretation of Local Basis provides an evaluation scheme for the global disentanglement property through the global warpage of the latent manifold. Our contributions are as follows:

1. We propose Local Basis, a set of traversal directions that can reliably traverse without escaping from the latent space to prevent image collapse. The latent traversal along Local Basis corresponds to the local coordinate mesh of local-geometry-describing submanifold.
2. We show that Local Basis leads to stable variation and better semantic factorization than global approaches. This result verifies our hypothesis on the close relationship between the semantic and geometric properties of the latent space in GANs.
3. We propose Iterative Curve-Traversal method, which is a way to trace the latent space in the curved trajectory. The trajectory of the images with this method shows a more stable variation compared to the linear traversal.
4. We introduce the metrics on the Grassmannian manifold to analyze the global geometry of the latent space through Local Basis. Quantitative analysis demonstrates that the $\mathcal{W}$-space of StyleGAN2 is still globally warped. This result provides an explanation for the limited success of the global basis and proves the importance of local approaches.

## 2 RELATED WORK

**Style-based Generators.** In recent years, GANs equipped with style-based generators (Karras et al., 2019; 2020b) have shown state-of-the-art performance in high-fidelity image synthesis. The style-based generator consists of two parts: a mapping network and a synthesis network. The mapping network encodes the isotropic Gaussian noise $\mathbf{z} \in \mathcal{Z}$ to an intermediate latent vector $\mathbf{w} \in \mathcal{W}$. The synthesis network takes $\mathbf{w}$ and generates an image while controlling the style of the image through $\mathbf{w}$. Here, $\mathcal{W}$-space is well known for providing a better disentanglement property compared to $\mathcal{Z}$ (Karras et al., 2019). However, there is still a lack of understanding about the effect of latent perturbation in a specific direction on the output image.

**Latent Traversal for Image Manipulation.** The impressive success of GANs in producing high-quality images has led to various attempts to understand their latent space. Early approaches (Radford et al., 2016; Upchurch et al., 2017) show that vector arithmetic on the latent space for the semantics holds, and StyleGAN (Karras et al., 2019) shows that mixing two latent codes can achieve style transfer. Some studies have investigated the supervised learning of latent directions while assuming access to the semantic attributes of images (Goetschalckx et al., 2019; Jahanian et al., 2019; Shen et al., 2020; Yang et al., 2021; Abdal et al., 2021). In contrast to these supervised methods, some recent studies have suggested novel approaches that do not use the prior knowledge of training dataset, such as the labels of human facial attributes. In Voynov & Babenko (2020), an unsupervised optimization method is proposed to jointly learn a candidate matrix and a corresponding reconstructor, which identifies the semantic direction in the matrix. GANSpace (Härkönen et al., 2020) finds a global basis for $\mathcal{W}$ in StyleGAN using a PCA, enabling a fast image manipulation. SeFa (Shen & Zhou, 2021) focuses on the first weight parameter right after the latent code, suggesting that it contains essential knowledge of an image variation. SeFa proposes singular vectors of the first weight parameter as meaningful global latent directions. StyleCLIP (Patashnik et al., 2021) achieves a state-of-the-art performance in the text-driven image manipulation of StyleGAN. StyleCLIP introduces an additional training to minimize the CLIP loss (Radford et al., 2021).

**Jacobian Decomposition.** Some works use the Jacobian matrix to analyze the latent space of GAN (Zhu et al., 2021; Wang & Ponce, 2021; Chiu et al., 2020; Ramesh et al., 2018). However, these methods focus on the Jacobian of the entire model, from the input noise $\mathbf{z}$ to the output image. Ramesh et al. (2018) suggested the right singular vectors of the Jacobian as local disentangled direc-

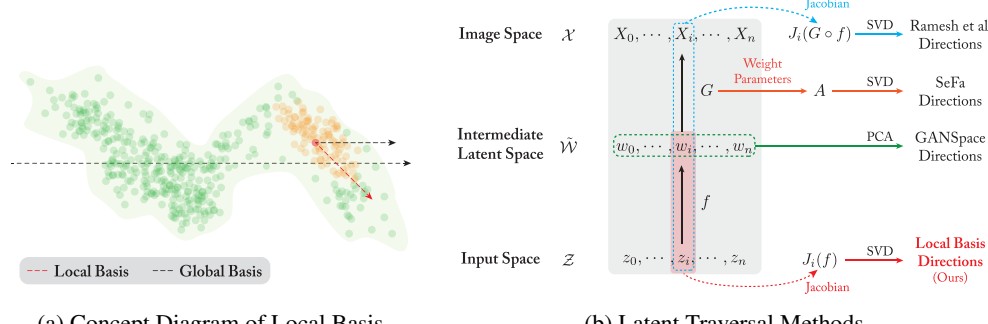

(a) Concept Diagram of Local Basis       (b) Latent Traversal Methods

Figure 1: **(a) Concept diagram of Local Basis**. The global basis reflects the global variation of latent space. Hence, traversing along the global basis may result in the escape from the latent space (shaded region). On the other hand, Local Basis closely follows the latent space. **(b) Comparison of Latent Traversal Methods** (Global methods: GANSpace (Härkönen et al., 2020) and SeFa (Shen & Zhou, 2021), Local methods: Ramesh et al. (2018) and *Local Basis* (Ours))

tions in the $\mathcal{Z}$ space. Zhu et al. (2021) proposed a latent perturbation vector that can change only a particular area of the image. The perturbation vector is discovered by taking the principal vector of the Jacobian to the target area and projecting it into the null space of the Jacobian to the complementary region. On the other hand, our Local Basis utilizes the Jacobian matrix of the partial network, from the input noise $z$ to the intermediate latent code $\mathbf{w}$, and investigates the black-box intermediate latent space from it. The top-$k$ Local Basis corresponds to the best-local-geometry-describing submanifolds. This intuition leads to exploiting Local Basis to assess the global geometry of the intermediate latent space.

## 3    TRAVERSING A CURVED LATENT SPACE

In this section, we introduce a method for finding a local-geometry-aware traversal direction in the intermediate latent space $\mathcal{W}$. The traversal direction is refered to as the *Local Basis* at $\mathbf{w} \in \mathcal{W}$. In addition, we evaluate the proposed Local Basis by observing how the generated image changes as we traverse the intermediate latent variable. Throughout this paper, we assess Local Basis of the $\mathcal{W}$-space in StyleGAN2 (Karras et al., 2020b). However, our methodology is not limited to StyleGAN2. See appendix for the results on StyleGAN (Karras et al., 2019) and BigGAN (Brock et al., 2018).

### 3.1    FINDING A LOCAL BASIS

Given a pretrained GAN model $M : \mathcal{Z} \to \mathcal{X}$, from the input noise space $\mathcal{Z}$ to the image space $\mathcal{X}$, we choose the intermediate layer $\tilde{\mathcal{W}}$ to discover Local Basis. We refer to the former part of the GAN model as the *mapping network* $f : \mathcal{Z} \to \tilde{\mathcal{W}}$. The image of the mapping network is denoted as $\mathcal{W} = f(\mathcal{Z}) \subset \tilde{\mathcal{W}}$. The latter part, a non-linear mapping from $\tilde{\mathcal{W}}$ to the image space $\mathcal{X}$, is denoted by $G : \tilde{\mathcal{W}} \to \mathcal{X}$. Local Basis at $\mathbf{w} \in \mathcal{W}$ is defined as the basis of the tangent space $T_{\mathbf{w}}\mathcal{W}$. This basis can be interpreted as a local-geometry-aware linear traversal direction starting from $\mathbf{w}$.

To define the tangent space of the intermediate latent space $\mathcal{W}$ properly, we assume that $\mathcal{W}$ is a *differentiable manifold*. Note that the support of the isotropic Gaussian prior $\mathcal{Z} = \mathbb{R}^{d_{\mathcal{Z}}}$ and the ambient space $\tilde{\mathcal{W}} = \mathbb{R}^{d_{\tilde{w}}}$ are already differentiable manifolds. The tangent space at $\mathbf{w}$, denoted by $T_{\mathbf{w}}\mathcal{W}$, is a vector space consisting of tangent vectors of curves passing through point $\mathbf{w}$. Explicitly,

$$T_{\mathbf{w}}\mathcal{W} = \{\, \dot{\gamma}(0) \mid \gamma : (-\epsilon, \epsilon) \to \mathcal{W}, \, \gamma(0) = \mathbf{w}, \text{ for } \epsilon > 0 \}. \tag{1}$$

Then, the differentiable mapping network $f$ gives a linear map $df_{\mathbf{z}}$ between the two tangent spaces $T_{\mathbf{z}}\mathcal{Z}$ and $T_{\mathbf{w}}\tilde{\mathcal{W}}$ where $\mathbf{w} = f(\mathbf{z})$.

$$df_{\mathbf{z}} : T_{\mathbf{z}}\mathcal{Z} \longrightarrow T_{\mathbf{w}}\mathcal{W} \longhookrightarrow T_{\mathbf{w}}\tilde{\mathcal{W}}, \quad \dot{\gamma}(0) \longmapsto (f \circ \gamma)^{\cdot}(0) \tag{2}$$

We utilize the linear map $df_{\mathbf{z}}$, called the *differential* of $f$ at $\mathbf{z}$, to find the basis of $T_{\mathbf{w}}\mathcal{W}$. Based on the manifold hypothesis in representation learning, we posit that the latent space of the image

space $\mathcal{X}$ in $\tilde{\mathcal{W}}$ is a lower-dimensional manifold embedded in $\mathcal{W}$. In this approach, we estimate the latent manifold as a lower-dimensional approximation of $\mathcal{W}$ describing its principal variations. The approximation manifold can be obtained by solving the low-rank approximation problem of $df_{\mathbf{z}}$. The manifold hypothesis is supported by the empirical distribution of singular values $\sigma_i^{\mathbf{z}}$. The analysis is provided in Fig 9 in the appendix.

The low-rank approximation problem has an analytic solution defined by Singular Value Decomposition (SVD). Because the matrix representation of $df_{\mathbf{z}}$ is a Jacobian matrix $(\nabla_{\mathbf{z}} f)(\mathbf{z}) \in \mathbb{R}^{d_{\tilde{\mathcal{W}}} \times d_{\mathcal{Z}}}$, Local Basis is obtained as the following: For the $i$-th right singular vector $\mathbf{u}_i^{\mathbf{z}} \in \mathbb{R}^{d_{\mathcal{Z}}}$, $i$-th left singular vector $\mathbf{v}_i^{\mathbf{w}} \in \mathbb{R}^{d_{\tilde{\mathcal{W}}}}$, and $i$-th singular value $\sigma_i^{\mathbf{z}} \in \mathbb{R}$ of $(\nabla_{\mathbf{z}} f)(\mathbf{z})$ with $\sigma_1^{\mathbf{z}} \geq \cdots \geq \sigma_n^{\mathbf{z}}$,

$$df_{\mathbf{z}}(\mathbf{u}_i^{\mathbf{z}}) = \sigma_i^{\mathbf{z}} \cdot \mathbf{v}_i^{\mathbf{w}} \text{ for } \forall i, \tag{3}$$

$$\text{Local Basis}(\mathbf{w} = f(\mathbf{z})) = \{\mathbf{v}_i^{\mathbf{w}}\}_{1 \leq i \leq n}. \tag{4}$$

Then, the $k$-dimensional approximation of $\mathcal{W}$ around $\mathbf{w}$ is described as the following because $\mathcal{Z} = \mathbb{R}^{d_{\mathcal{Z}}}$ (if $\sigma_k^{\mathbf{z}} > 0$). Note that $\mathcal{W}_{\mathbf{w}}^k$ is a submanifold[1] of $\mathcal{W}$ corresponding to the $k$ components of Local Basis, i.e. $T_{\mathbf{w}} \mathcal{W}_{\mathbf{w}}^k = \text{span}\{\mathbf{v}_i^{\mathbf{w}} : 1 \leq i \leq k\}$.

$$\mathcal{W}_{\mathbf{w}}^k = \left\{ f\left(\mathbf{z} + \sum_i t_i \cdot \mathbf{u}_i^{\mathbf{z}}\right) \mid t_i \in (-\epsilon_i, \epsilon_i), \text{ for } 1 \leq i \leq k \right\} \tag{5}$$

**Locally affine mapping network**   In this paragraph, we focus on the locally affine mapping network $f$, which is one of the most widely adopted GAN structures, such as MLP or CNN layers with ReLU or leaky-ReLU activation functions. This type of mapping network has several well-suited properties for Local Basis.

$$f(\mathbf{z}) = \sum_{p \in \Omega} \mathbf{1}_{\mathbf{z} \in p} \left(\mathbf{A}_p \mathbf{z} + \mathbf{b}_p\right) \tag{6}$$

where $\Omega$ denotes a partition of $\mathcal{Z}$, and $\mathbf{A}_p$ and $\mathbf{b}_p$ are the parameters of the local affine map. With this type of mapping network $f$, it is clear that the intermediate latent space $\mathcal{W}$ satisfies a differentiable manifold property at least locally on the interior of each $p \in \Omega$. The region where the property may not hold, the intersection of several closure of $p$'s in $\Omega$, has measure zero in $\mathcal{Z}$.

Moreover, the Jacobian matrix $(\nabla_{\mathbf{z}} f)(\mathbf{z})$ becomes a locally constant matrix. Then, the approximating manifold $\mathcal{W}_{\mathbf{w}}^k$ (Eq 5) satisfies the submanifold condition, and is consistent locally for each $p$, avoiding being defined for each $\mathbf{w}$. In addition, the linear traversal of the latent variable $\mathbf{w}$ along $\mathbf{v}_i^{\mathbf{w}}$ can be described as the curve on $\mathcal{W}$ (Eq 7). Most importantly, these curves on $\mathcal{W}$ (Eq 7), starting from $\mathbf{w}$ in the direction of Local Basis, corresponds to the local coordinate mesh of $\mathcal{W}_{\mathbf{w}}^k$.

$$\text{Traversal}(\mathbf{w} = f(\mathbf{z}), \mathbf{v}_i^{\mathbf{w}}) : (-\epsilon, \epsilon) \longrightarrow \mathcal{Z} \xrightarrow{f} \mathcal{W}, \quad t \mapsto \left(\mathbf{z} + \frac{t}{\sigma_i^{\mathbf{z}}} \cdot \mathbf{u}_i^{\mathbf{z}}\right) \mapsto (\mathbf{w} + t \cdot \mathbf{v}_i^{\mathbf{w}}) \tag{7}$$

**Equivalence to Local PCA**   To provide additional intuition about Local Basis, we prove the following proposition. The proposition shows that Local Basis is equivalent to applying a PCA on the samples on $\mathcal{W}$ around $\mathbf{w}$.

**Proposition 1** (Equivalence to Local PCA). *Consider the **Local PCA** problem around the base latent variable $\mathbf{w}_b = f(\mathbf{z}_b)$ on $\mathcal{W}$, i.e. PCA of the latent variable samples $\mathbf{w}'$ around $\mathbf{w}_b$.*

$$\mathbf{w}' = T_1 f(\mathbf{z}_b + c \cdot \epsilon) \quad \text{with } \epsilon \sim N(0, I) \text{ and for some small } c > 0. \tag{8}$$

*where $T_1 f(\mathbf{z}) = \mathbf{w}_b + (\nabla_{\mathbf{z}_b} f)(\mathbf{z} - \mathbf{z}_b)$ is the linear approximation of $f$ around $\mathbf{z}_b$. Then, the principal components discovered in the Local PCA problem are equivalent to Local Basis at $\mathbf{w}_b$.*

## 3.2   Iterative Curve-Traversal

We suggest a natural curve-traversal that can keep track of the $\mathcal{W}$-manifold and an iterative method to implement it. We divide the long curved trajectory into small pieces and approximate each piece

---

[1]Strictly speaking, $\mathcal{W}_{\mathbf{w}}^k$ may not satisfy the conditions of the submanifold. The injectivity of $df_{\mathbf{z}}$ on the domain $\{\mathbf{z} + \sum_i t_i \cdot \mathbf{u}_i^{\mathbf{z}} \mid t_i \in (-\epsilon_i, \epsilon_i), \text{ for } 1 \leq i \leq k\}$ is a sufficient condition for the submanifold. As described below, this sufficient condition is satisfied under the locally affine mapping network $f$ and $\sigma_k^{\mathbf{z}} > 0$.

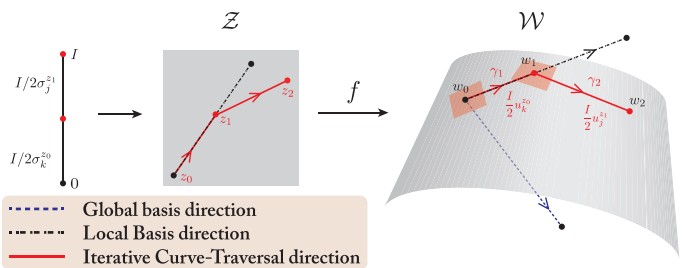

Figure 2: **Illustration of Iterative Curve-Traversal** (for $N = 2$).

by the local curves using Local Basis. We call this curve-traversal *Iterative Curve-Traversal* method. Consistent with the linear traversal method, we consider Iterative Curve-Traversal $\gamma$ departing in the direction of a Local Basis. Explicitly, for a sufficiently large $c > 0$,

$$\gamma : (-c, c) \longrightarrow \mathcal{W}, \quad \gamma(0) = \mathbf{w}, \; \dot{\gamma}(0) = \mathbf{v}_k^{\mathbf{w}} \quad \text{for some } 1 \leq k \leq d_{\mathcal{W}} \tag{9}$$

where $\{\mathbf{v}_i^{\mathbf{w}}\}_i = \text{Local Basis}(\mathbf{w})$. We split the curve-traversal $\gamma$ into $N$ pieces $\gamma_n$ and denote each $n$-th iterate in $\mathcal{W}$ and $\mathcal{Z}$ as $\mathbf{w}_n$ and $\mathbf{z}_n$ for $1 \leq n \leq N$. The starting point of the traversal is denoted as the 0-th iterate $\mathbf{w} = \mathbf{w}_0$, $\mathbf{z} = \mathbf{z}_0$, and $\mathbf{w}_0 = f(\mathbf{z}_0)$. (Fig 2) Note that to find Local Basis at $\mathbf{w}_n$, we need a corresponding $\mathbf{z}_n \in \mathcal{Z}$ such that $\mathbf{w}_n = f(\mathbf{z}_n)$.

Below, we describe the positive part $\gamma^+ = \gamma|_{[0,c)}$ of Iterative Curve-Traversal. For the negative part, we repeat the same procedure using the reversed tangent vector $-\mathbf{v}_k^{\mathbf{w}}$. The first step $\gamma_1^+$ of Iterative Curve-Traversal method with perturbation intensity $I$ is as follows:

$$\gamma_1^+ : [0, I/(N \cdot \sigma_k^{\mathbf{z}_0})] \longrightarrow \mathcal{Z} \xrightarrow{f} \mathcal{W}, \quad t \longmapsto (\mathbf{z}_0 + t \cdot \mathbf{u}_k^{\mathbf{z}_0}) \longmapsto f(\mathbf{z}_0 + t \cdot \mathbf{u}_k^{\mathbf{z}_0}) \tag{10}$$

$$\mathbf{z}_1 = \mathbf{z}_0 + \frac{I}{(N \cdot \sigma_k^{\mathbf{z}_0})} \mathbf{u}_k^{\mathbf{z}_0}, \quad \mathbf{w}_1 = f(\mathbf{z}_1) \tag{11}$$

Note that $\mathbf{w}_1$ is the endpoint of the curve $\gamma_1^+$ and $\dot{\gamma}_1^+(0) = \mathbf{v}_k^{\mathbf{w}}$. We scale the step size in $\mathcal{Z}$ by $1/\sigma_k^{\mathbf{z}_0}$ to ensure each piece of curve has a similar length of $(I/N)$. To preserve the variation in semantics during the traversal, the departure direction of $\gamma_2^+$ is determined by comparing the similarity between the previous departure direction $\mathbf{v}_k^{\mathbf{w}_0}$ and Local Basis at $\mathbf{w}_1$. The above process is repeated $N$-times. (The algorithm for Iterative Curve-Traversal can be found in the appendix.)

$$\dot{\gamma}_2^+(0) = \mathbf{v}_j^{\mathbf{w}_1} \quad \text{where} \quad j = \underset{1 \leq i \leq d_{\mathcal{W}}}{\operatorname{argmax}} |\langle \mathbf{v}_k^{\mathbf{w}_0}, \mathbf{v}_i^{\mathbf{w}_1} \rangle| \tag{12}$$

### 3.3 Results of Local Basis traversal

We evaluate Local Basis by observing how the generated image changes as we traverse $\mathcal{W}$-space in StyleGAN2 and by measuring FID score for each perturbation intensity. The evaluation is based on two criteria: Robustness and Semantic Factorization.

**Robustness** Fig 3 and 4 present the Robustness Test results[2]. In Fig 3, the traversal image of Local Basis is compared with those of the global methods (GANSpace (Härkönen et al., 2020) and SeFa (Shen & Zhou, 2021)) under the strong perturbation intensity of 12 along the 1st and 2nd direction of each method. The perturbation intensity is defined as the traversal path length in $\mathcal{W}$. The two global methods show severe degradation of the image compared to Local Basis. Moreover, we perform a quantitative assessment of robustness. We measure the FID score for 10,000 traversed images for each perturbation intensity. In Fig 4, the global methods show the relatively small FID under the small perturbation. But, as we impose the stronger perturbation, the FID scores on the global methods increase sharply, implying the image collapse in Fig 3. By contrast, Local Basis achieves much smaller FID scores with and without Iterative Curve-Traversal.

---

[2]Ramesh et al. (2018) is not compared because it took hours to get a traversal direction of an image. See appendix for the Qualitative Robustness Test results of Ramesh et al. (2018).

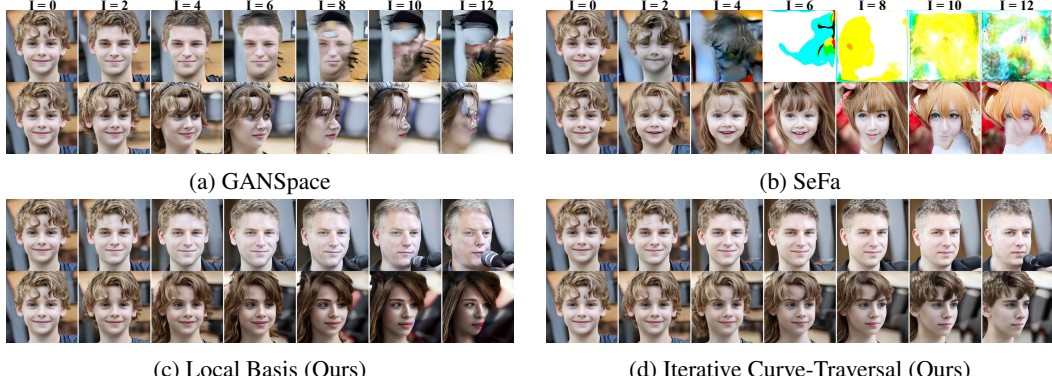

(a) GANSpace          (b) SeFa

(c) Local Basis (Ours)        (d) Iterative Curve-Traversal (Ours)

Figure 3: **Qualitative Robustness Test** on the $\mathcal{W}$-space of the StyleGAN2 (Karras et al., 2020b) trained on FFHQ. Each traversal image is generated by the linear traversal on $\mathcal{W}$ except for (d) under the strong perturbation intensity $I$ of up to 12. The intensity is linearly increased from 0 to 12 for each column. We infer the deterioration of the traversal image along the global method is due to the escape of the latent traversal from the latent manifold. (See the appendix for the additional Robustness Test results along the first 10 components of Local Basis.)

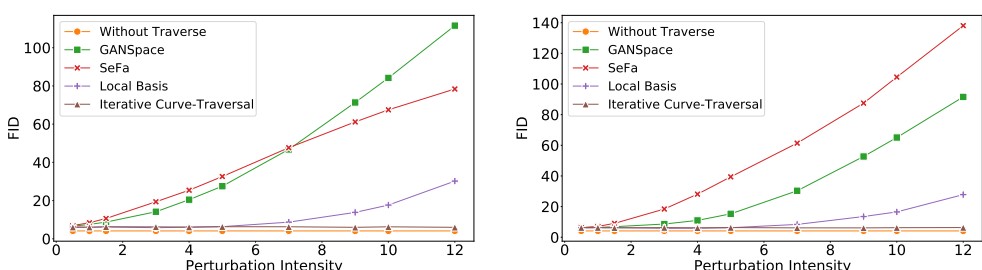

Figure 4: **Quantitative Robustness Test** on the $\mathcal{W}$-space of the StyleGAN2 (Karras et al., 2020b) trained on FFHQ. Fréchet Inception Distance (FID) (Heusel et al., 2017) is measured for 10,000 traversed images for each perturbation intensity. **Left**: 1st direction, **Right**: 2nd direction

We interpret the degradation of image as due to the deviation of trajectory from $\mathcal{W}$. The theoretical interpretation shows that the linear traversal along Local Basis corresponds to a local coordinate axis on $\mathcal{W}$, at least locally. Therefore, the traversal along Local Basis is guaranteed to stay close to $\mathcal{W}$ even under the longer traversal. However, we cannot expect the same property on the global basis because it is based on the global geometry. Iterative Curve-Traversal shows more stable traversal because of its stronger tracing to the latent manifold. This further supports our interpretation.

**Semantic Factorization** Local Basis is discovered in terms of singular vectors of $df_{\mathbf{z}}$. The disentangled correspondence, between Local Basis and the corresponding singular vectors in the prior space, induces a semantic-factorization in Local Basis. Fig 5 and 6 presents the semantics of the image discovered by Local Basis. In Fig 5, we compare the semantic factorizations of Local Basis and GANSpace (Härkönen et al., 2020) for the particular semantics discovered by GANSpace. For each interpretable traversal direction of GANSpace provided by the authors, the corresponding Local Basis is chosen by the one with the highest cosine similarity. For a fair comparison, each traversal is applied to the specific subset of layers in the synthesis network (Karras et al., 2020b) provided by the authors of GANSpace with the same perturbation intensity. In particular, as we impose the stronger perturbation (from left to right), GANSpace shows the image collapse in Fig 5a and entanglement of semantics (*Glasses + Head Raising*) in Fig 5d. However, Local Basis does not show any of those problems. Fig 6 provides additional examples of semantic factorization where the latent traversal is applied to a subset of layers predefined in StyleGAN. The subset of the layers is selected as one of four, i.e. *coarse*, *middle*, *fine*, or *all* styles. Local Basis shows decent factorization of semantics such as Body Length of car and Age of cat in LSUN (Yu et al., 2015) in Fig 6.

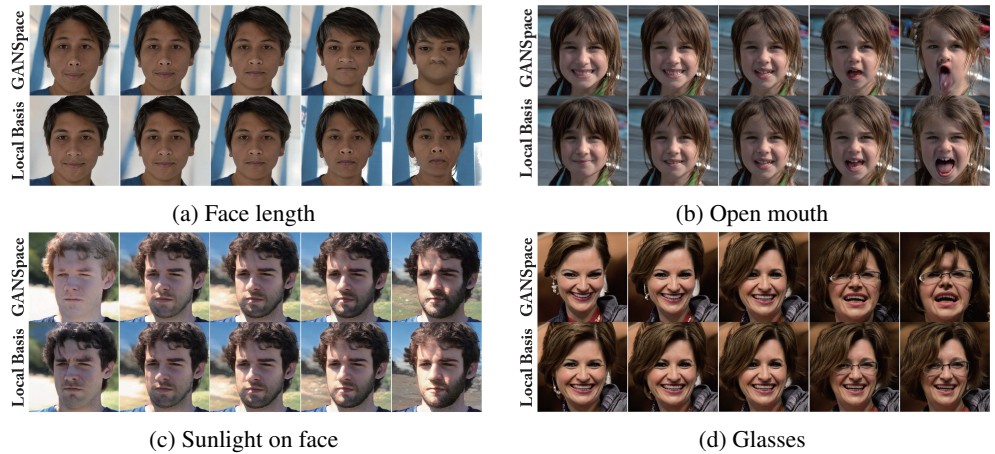

Figure 5: **Comparison of Semantic Factorization** between Local Basis and GANSpace on pre-trained StyleGAN2-FFHQ. We compare the semantic-factorizing directions of GANSpace provided by the authors (Härkönen et al., 2020) with Local Basis of the highest cosine similarity. Local Basis factorizes semantics of image better, notably without collapsing compared to the GANSpace.

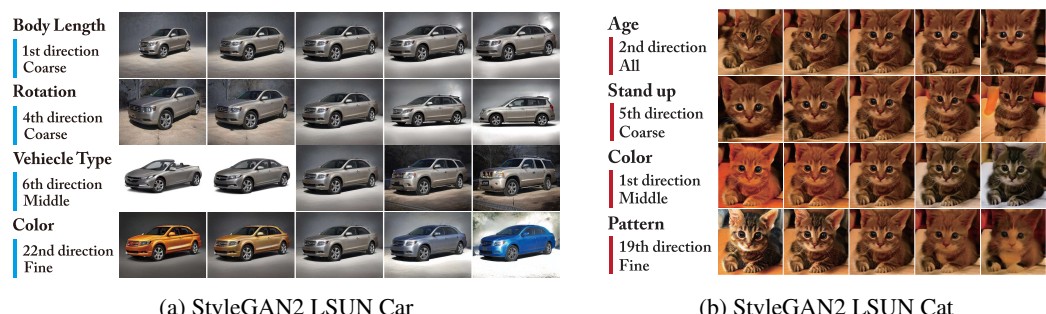

Figure 6: **Additional Semantic Factorization examples** of Local Basis. The examples are discovered by manual inspection due to the unsupervised nature while applying the latent traversal to a subset of the layers predefined in StyleGAN (Karras et al., 2019): *coarse, middle, fine*, and *all* styles. (See the appendix for the additional examples of Semantic Factorization without layer restriction.)

### 3.4 EXPLORATION INSIDE ABSTRACT SEMANTICS

Abstract semantics of image often consists of several lower-level semantics. For instance, *Old* can be represented as the correlated distribution of *Hair color*, *Wrinkle*, *Face length*, etc. In this section, we show that the adaptation of Iterative Curve-Traversal can explore the variation of abstract semantics, which is represented by the cone-shaped region of the generative factors (Träuble et al., 2021).

Because of its text-driven nature, we utilize the global basis[3] from StyleCLIP (Patashnik et al., 2021) corresponding to the abstract semantics of *Old*. Then, we consider the modification of Iterative Curve-Traversal following the given global basis $\mathbf{v}_{global}$. To be more specific, the departure direction of each piece of curve $\gamma_i$ in Eq 12 is chosen by the similarity to $\mathbf{v}_{global}$, not by the similarity to previous departure direction. The results for *old* are provided in Fig 7. (See the appendix for other examples.) *Step size* denotes the length of each piece of curve, i.e. $(I/N)$ in Sec 3.2. For a fair comparison, the overall perturbation intensity $I$ is fixed to 4 by adjusting the number of steps $N$. The linear traversal along the global basis adds only wrinkles to the image and the image collapses shortly. On the contrary, both Iterative Curve-Traversal methods obtain the diverse and high-fidelity image manipulation for the target semantics *old*. In particular, the diversity is greatly increased as we add stochasticity to the step size. We interpret this diversity as a result of the increased exploration area from the stochastic step size while exploiting the high-fidelity of Iterative Curve-Traversal.

---

[3]We use the global basis defined on $\mathcal{W}^+$ (Tov et al., 2021). See the appendix for detail.

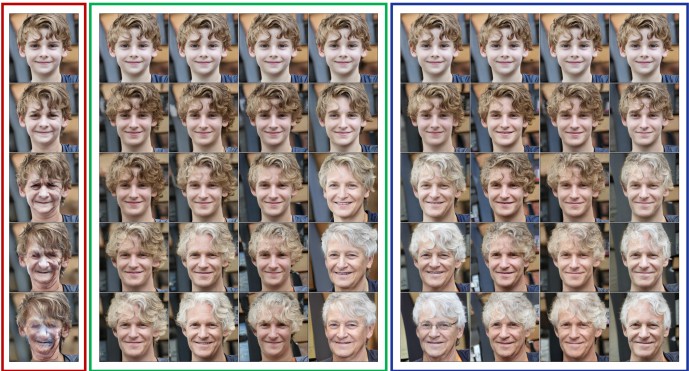

Figure 7: **Iterative Curve-Traversal guided by global basis** from StyleCLIP for the semantics of *old*. **Left**: Linear traversal along global basis. **Middle**: Iterative Curve-Traversal of fixed step size (Stepsize = (0.02, 0.04, 0.08, 0.16)). **Right**: Stochastic Iterative Curve-Traversal (Step size is sampled from Uniform Noise on $[0.05, 0.15]$)

## 4    EVALUATING WARPAGE OF $\mathcal{W}$-MANIFOLD

In this section, we provide an explanation for the limited success of the global basis in $\mathcal{W}$-space of StyleGAN2. In Sec 3, we showed that Local Basis corresponds to the generative factors of data. Hence, the linear subspace spanned by Local Basis, which is the tangent space $T_{\mathbf{w}}\mathcal{W}^k_{\mathbf{w}}$ in Eq 5, describes the local principal variation of image. In this regard, we assess the global disentanglement property by evaluating the consistency of the tangent space at each $\mathbf{w} \in \mathcal{W}$. We refer to the inconsistency of the tangent space as the warpage of the latent manifold. Our evaluation proves that $\mathcal{W}$-manifold is warped globally. In this section, we present the quantitative evaluation of the global warpage by introducing the Grassmannian Metric. The qualitative evaluation by observing the subspace traversal is provided in the appendix. The subspace traversal denotes a simultaneous traversal in multiple directions.

**Grassmannian Manifold**    Let $V$ be the vector space. The Grassmannian manifold $\mathrm{Gr}(k, V)$ (Boothby, 1986) is defined as the set of all $k$-dimensional linear subspaces of $V$. We evaluate the global warpage of $\mathcal{W}$-manifold by measuring the Grassmannian distance between the linear subspaces spanned by top-$k$ Local Basis of each $\mathbf{w} \in \mathcal{W}$. The reason for measuring the distance for top-$k$ Local Basis is the manifold hypothesis. The linear subspace spanned by top-$k$ Local Basis corresponds to the tangent space of the $k$-dimensional approximation of $\mathcal{W}$ (Eq 5). From this perspective, a large Grassmannian distance means that the $k$-dimensional local approximation of $\mathcal{W}$ severely changes. Likewise, we consider the subspace spanned by the top-$k$ components for the global basis. In this study, two types of metrics (i.e. Projection metric and Geodesic metric) are adopted as metrics of the Grassmannian manifold.

**Grassmannian Metric**    First, for two subspaces $W, W' \in \mathrm{Gr}(k, V)$, let the projection into each subspace be $P_W$ and $P_{W'}$, respectively. Then the **Projection Metric** (Karrasch, 2017) on $\mathrm{Gr}(k, V)$ is defined as follows.

$$d_{\mathrm{proj}}(W, W') = \|P_W - P_{W'}\| \tag{13}$$

where $\|\cdot\|$ denotes the operator norm.

Second, let $M_W, M_{W'} \in \mathbb{R}^{d_V \times k}$ be the column-wise orthonormal matrix of which columns span $W, W' \in \mathrm{Gr}(k, V)$, respectively. Then, the **Geodesic Metric** (Ye & Lim, 2016) on $\mathrm{Gr}(k, V)$, which is induced by canonical Riemannian structure, is formulated as follows.

$$d_{\mathrm{geo}}(W, W') = \left(\sum_{i=1}^{k} \theta_i^2\right)^{1/2} \tag{14}$$

where $\theta_i = \cos^{-1}(\sigma_i(M_W^\top M_{W'}))$ denotes the $i$-th principal angle between $W$ and $W'$.

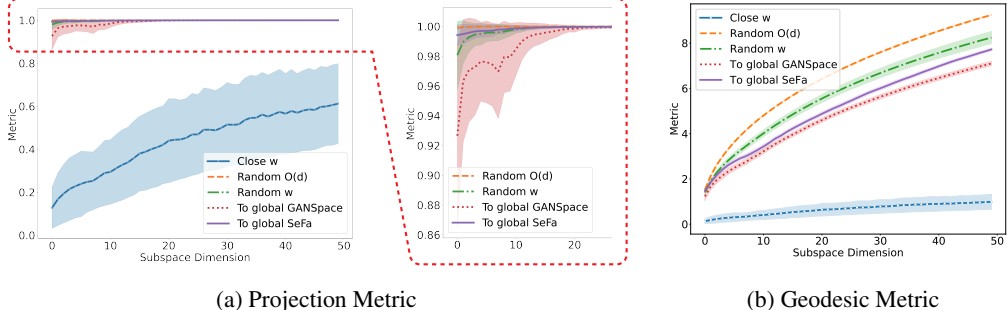

(a) Projection Metric                             (b) Geodesic Metric

Figure 8: **Grassmannian metric.** The shaded region illustrates (mean $\pm$ standard deviation) intervals of each score. Above all, *Random* $\mathbf{w}$ metric is much larger than the *Close* $\mathbf{w}$. This means a large variation of Local Basis on $\mathcal{W}$, which demonstrates that $\mathcal{W}$-space is globally warped. Moreover, the metric result shows the local consistency of Local Basis and the existence of limited global alignment on $\mathcal{W}$. (See Sec 4 for detail)

**Evaluation**    We evaluate the global warpage of the $\mathcal{W}$-manifold by comparing the five distances as we vary the subspace dimension.

1. **Random $O(d)$**: Between two random basis of $\mathbb{R}^{d_{\mathcal{W}}}$ uniformly sampled from $O(d_{\mathcal{W}})$
2. **Random $\mathbf{w}$**: Between two Local Basis from two random $\mathbf{w} \in \mathcal{W}$
3. **Close $\mathbf{w}$**: Between two Local Basis from two close $\mathbf{w}', \mathbf{w} \in \mathcal{W}$ (See appendix for the Grassmannian metric with various $\epsilon = |\mathbf{z}' - \mathbf{z}|$.)

$$\mathbf{w}' = f(\mathbf{z}'), \quad \mathbf{w} = f(\mathbf{z}) \qquad \text{where } |\mathbf{z}' - \mathbf{z}| = 0.1 \tag{15}$$

4. **To global GANSpace**: Between Local Basis and the global basis from GANSpace
5. **To global SeFa**: Between Local Basis and the global basis from SeFa

Fig 8 shows the above five Grassmannian metrics. We report the metric results from 100 samples for the *Random $O(d_{\mathcal{W}})$* and 1,000 samples for the others. The Projection metric increases in order of *Close $\mathbf{w}$, To global GANSpace, Random $\mathbf{w}$, To global SeFa, and Random $O(d_{\mathcal{W}})$*. For the Geodesic metric, the order is reversed for *Random $\mathbf{w}$ and To global SeFa*. Most importantly, the *Random $\mathbf{w}$* metric is much larger than *Close $\mathbf{w}$*. This shows that there is a large variation of Local Basis on $\mathcal{W}$, which proves that $\mathcal{W}$-space is globally warped. In addition, *Close $\mathbf{w}$* metric is always significantly smaller than the others, which implies the local consistency of Local Basis on $\mathcal{W}$. Finally, the metric results prove the existence of limited global disentanglement on $\mathcal{W}$. *Random $\mathbf{w}$* is smaller than *Random $O(d)$*. This order shows that Local Basis on $\mathcal{W}$ is not completely random, which implies the existence of a global alignment. In this regard, both *To global* results prove that the global basis finds the global alignment to a certain degree. *To global GANSpace* lies in between *Close $\mathbf{w}$* and *Random $\mathbf{w}$*. *To global SeFa* does so on the Geodesic metric and is similar to *Random $\mathbf{w}$* on the Projection metric. However, the large gap between *Close $\mathbf{w}$* and both *To global* implies that the discovered global alignment is limited.

## 5   CONCLUSION

In this work, we proposed a method for finding a meaningful traversal direction based on the local-geometry of the intermediate latent space of GANs, called Local Basis. Motivated by the theoretical explanation of Local Basis, we suggest experiments to evaluate the global geometry of the latent space and an iterative traversal method that can trace the latent space. The experimental results demonstrate that Local Basis factorizes the semantics of images and provides a more stable transformation of images with and without the proposed iterative traversal. Moreover, the suggested evaluation of the $\mathcal{W}$-space in StyleGAN2 proves that the $\mathcal{W}$-space is globally distorted. Therefore, the global method can find a limited global consistency from $\mathcal{W}$-space.

ACKNOWLEDGEMENT

This work was supported by the NRF grant [2021R1A2C3010887], the ICT R&D program of MSIT/IITP [2021-0-00077] and MOTIE [P0014715].

ETHICS STATEMENT

The limitations and the potential negative societal impacts of our work are that Local Basis would reflect the bias of data. The GANs learn the probability distribution of data through samples from it. Thus, unlike the likelihood-based method such as Variational Autoencoder (Kingma & Welling, 2014) and Flow-based models (Kingma & Dhariwal, 2018), the GANs are more likely to amplify the dependence between the semantics of data, even the bias of it. Because Local Basis finds a meaningful traversal direction based on the local-geometry of latent space, Local Basis would show the bias of data as it is. Moreover, if Local Basis is applied to real-world problems like editing images, Local Basis may amplify the bias of society. However, in order to fix a problem, we have to find a method to analyze it. In this respect, Local Basis can serve as a tool to analyze the bias.

REPRODUCIBILITY STATEMENT

To ensure the reproducibility of this study, we attached the entire source code in the supplementary material. Every figure can be reproduced by running the jupyter notebooks in notebooks/*. In addition, the proof of Proposition 1 is included in the appendix.

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

# A PROPOSITION PROOF

Denote $(\nabla_{\mathbf{z}_b} f)$ by $J$. Then, from $\mathbf{w}' = T_1 f(\mathbf{z}_b + c \cdot \epsilon)$,

$$\mathbf{w}' = \mathbf{w}_b + c \cdot J\epsilon \sim N(\mathbf{w}_b, c^2 \cdot JJ^\mathsf{T}) \tag{16}$$

The first principal component $\mathbf{v}_1$ is the vector such that $\mathbf{v}_1^\mathsf{T}(c \cdot J\epsilon)$ has the maximum variance.

$$\mathrm{Var}(\mathbf{v}_1^\mathsf{T}(c \cdot J\epsilon)) = c^2 \cdot ||\mathbf{v}_1^\mathsf{T} J||_2^2 \tag{17}$$

Therefore,

$$\mathbf{v}_1 = \underset{||\mathbf{v}||_2=1}{\mathrm{argmax}}\, \mathrm{Var}(\mathbf{v}^\mathsf{T}(c \cdot J\epsilon)) = \underset{||\mathbf{v}||_2=1}{\mathrm{argmax}}\, ||J^\mathsf{T} \cdot \mathbf{v}||_2 \tag{18}$$

Clearly, $\mathbf{v}_1$ corresponds to the first right singular vector of $J^\mathsf{T}$, i.e. the first left singular vector of $J$, from the linear operator norm maximizing property of singular vectors. Inductively, $k$-th principal component $\mathbf{v}_k$ is the vector such that

$$\mathbf{v}_k = \underset{||\mathbf{v}||_2=1}{\mathrm{argmax}}\, \mathrm{Var}(\mathbf{v}^\mathsf{T}(c \cdot J\epsilon)) \quad \text{where} \quad \mathbf{v}_k \perp \{\mathbf{v}_1, \mathbf{v}_2, \cdots, \mathbf{v}_{k-1}\} \tag{19}$$

Thus, $\mathbf{v}_k$ becomes the $k$-th left singular vector of $J$. Therefore, the principal components from the Local PCA problem are equivalent to Local Basis at $\mathbf{w}_b$.

# B ALGORITHM

---

**Algorithm 1** Local Basis

---

**Input:**
 1: $z \in \mathbb{R}^{d_{\mathcal{Z}}}$ is the input code.
 2: $f : \mathbb{R}^{d_{\mathcal{Z}}} \to \mathbb{R}^{d_{\mathcal{W}}}$ is the mapping network.

**Output:** LOCALBASIS$(z, f)$
 3: $w \leftarrow f(z)$
 4: $J \in \mathbb{R}^{d_{\mathcal{W}} \times d_{\mathcal{Z}}} \leftarrow$ JACOBIAN$(z, w)$
 5: $U, S, V \leftarrow$ SVD$(J)$
 6: **return** $\{U, S, V\}$

---

---

**Algorithm 2** Iterative Curve-Traversal along positive direction

---

**Input:**
 1: $z \in \mathbb{R}^{d_{\mathcal{Z}}}$ is the input code.
 2: $f : \mathbb{R}^{d_{\mathcal{Z}}} \to \mathbb{R}^{d_{\mathcal{W}}}$ is the mapping network.
 3: $k \in [1, \min\{d_{\mathcal{Z}}, d_{\mathcal{W}}\}]$ is the ordinal number of direction to traverse.
 4: $I$ is the total perturbation intensity.
 5: $N \geq 1$ is the number of iterations.

**Output:** ITERATIVETRAVERSAL$(z, f, k, I, N)$
 6: $z_0 \leftarrow z$
 7: $c \leftarrow ones(d_{\mathcal{Z}}, 1)$
 8: **for** $i \in [0, N)$ **do**
 9: $\quad U, S, V \leftarrow$ LOCALBASIS$(z_i, f)$
 10: $\quad$ **if** i ¿ 0 **then**
 11: $\quad\quad c \leftarrow U^T \cdot u_{i-1}$
 12: $\quad\quad k \leftarrow \arg\max(|c|)$ $\quad\quad\quad \triangleright$ The row number most similar to the previously selected basis.
 13: $\quad$ **end if**
 14: $\quad u_i, v_i \leftarrow \mathrm{sign}(c_k)U_k, \mathrm{sign}(c_k)V_k$ $\quad\quad\quad\quad\quad \triangleright$ Aligns with previous orientation
 15: $\quad s_i \leftarrow S_{kk}$
 16: $\quad z_{i+1} \leftarrow z_i + \frac{I}{s_i \cdot N} v_i$
 17: **end for**
 18: **return** $\{z_0, ..., z_N\}$

---

## C  MODEL AND COMPUTATION RESOURCE DETAILS

**Model**    We evaluate GANSpace (Härkönen et al., 2020), SeFa (Shen & Zhou, 2021), and Local Basis on StyleGAN2 models for FFHQ (Karras et al., 2019) and LSUN (Yu et al., 2015) provided by the authors (Karras et al., 2020b).

**Computation Resource**    We generated Latent traversal results on the environment of TITAN RTX with Intel(R) Xeon(R) Gold 5220 CPU @ 2.20GHz. However, it requires a low computational cost to get a Local Basis. For example, on the environment of GTX 1660 with Ryzen 5 2600, computing a Local Basis takes about 0.05 seconds.

## D  CODE LICENSE

The files models/wrappers.py, notebooks/ganspace_utils.py and notebooks/notebook_utils.py are a derivative of the GANSpace, and are provided under the Apache 2.0 license. The directory netdissect is a derivative of the GAN Dissection project, and is provided under the MIT license. The directories models/biggan and models/stylegan2 are provided under the MIT license.

# E DISTRIBUTION OF SIGULAR VALUES OF JACOBIAN

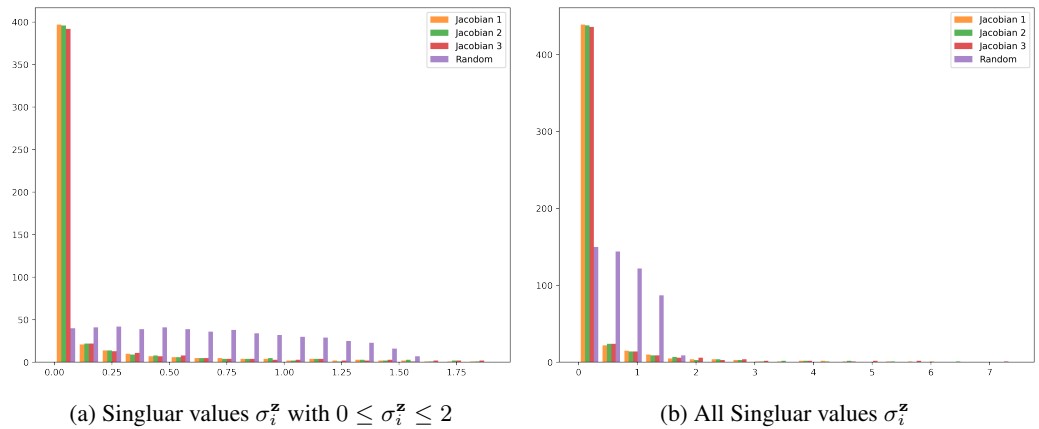

(a) Singluar values $\sigma_i^{\mathbf{z}}$ with $0 \leq \sigma_i^{\mathbf{z}} \leq 2$

(b) All Singluar values $\sigma_i^{\mathbf{z}}$

Figure 9: **Histogram of the Singular Values** $\sigma_i^{\mathbf{z}}$ of $df_{\mathbf{z}}$ for three random $\mathbf{z}$ and the random matrix. The random matrix is sampled from the Gaussian distribution, then transformed to have the mean and standard deviation of the 100 Jacobian matrix. The sharp peak around zero demonstrates that most of the linear perturbation from $\mathbf{z}$ collapses. This observation proves our manifold hypothesis. To better represent the sparsity of singular values, we provide the histogram of sigular values $\sigma_i^{\mathbf{z}}$ with $0 \leq \sigma_i^{\mathbf{z}} \leq 2$ separately.

# F GRASSMANNIAN METRIC

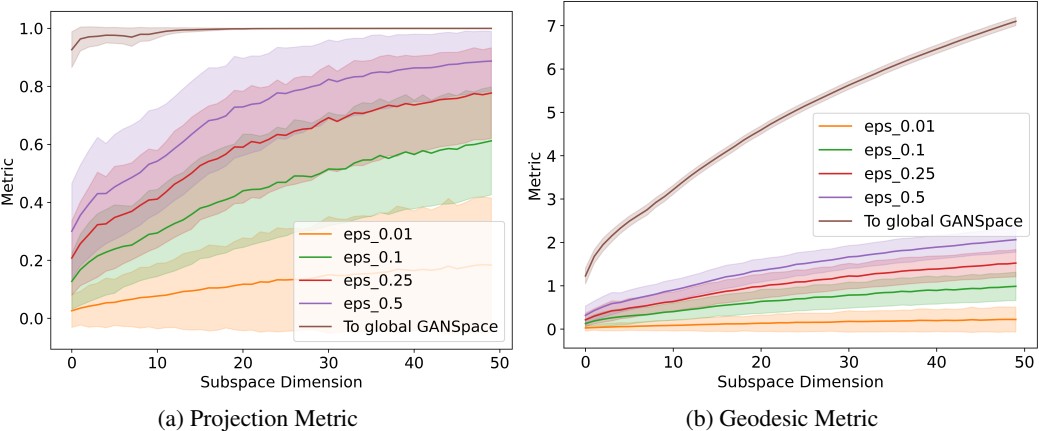

(a) Projection Metric

(b) Geodesic Metric

Figure 10: **Grassmannian metric between two close** $\mathbf{w}, \mathbf{w}' \in \mathcal{W}$ **as we vary** $\epsilon$. We denote $\epsilon = |\mathbf{z}' - \mathbf{z}|$ where $\mathbf{w}' = f(\mathbf{z}')$, $\mathbf{w} = f(\mathbf{z})$. As expected, the Grassmannian metric monotonically increases as we increase $\epsilon$. However, even for the case of $\epsilon = 0.5$, the evaluated metric is much smaller than *To global GANSpace*. Therefore, regardless of $\epsilon$, every metric for *Close* $\mathbf{w}$ supports our claim for the global warpage of $\mathcal{W}$-space. In the main text, we present only the case of $\epsilon = 0.1$. The reported Grassmannian metrics, Fig 10 in the supplementary material and Fig 8 in the main text, are evaluated on the SytleGAN2 model trained on FFHQ.

# G More Latent Traversal Examples

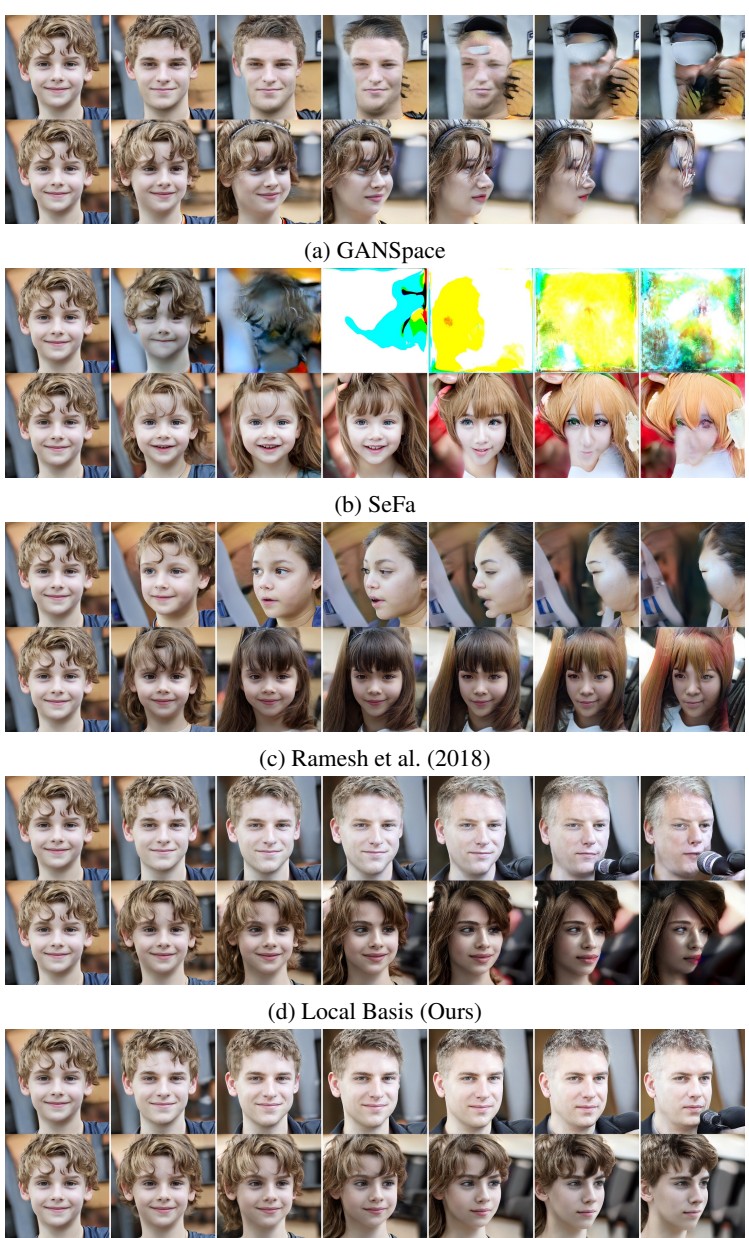

(a) GANSpace

(b) SeFa

(c) Ramesh et al. (2018)

(d) Local Basis (Ours)

(e) Iterative Curve-Traversal (Ours)

Figure 11: **Enlarged figure for Fig 3**. Each row represents latent traversal on the $\mathcal{W}$-space of the StyleGAN2-FFHQ, except for (c). Ramesh et al. (2018) provides the local traversal directions on $\mathcal{Z}$. Except for (e), each traversal image is generated by the linear traversal. The latent code $\mathbf{w}$ is perturbed up to 12 along the 1st and 2nd direction of the corresponding method. The perturbation intensity is linearly increased from 0 to 12 for each column. Since Ramesh et al. (2018) is defined on $\mathcal{Z}$, we downscaled the perturbation intensity by the singular values from Local Basis for a fair comparison. In the case of the existing methods, the quality of the image gets severely degraded as we perturb stronger. On the other hand, Local Basis shows a relatively stable traversal.

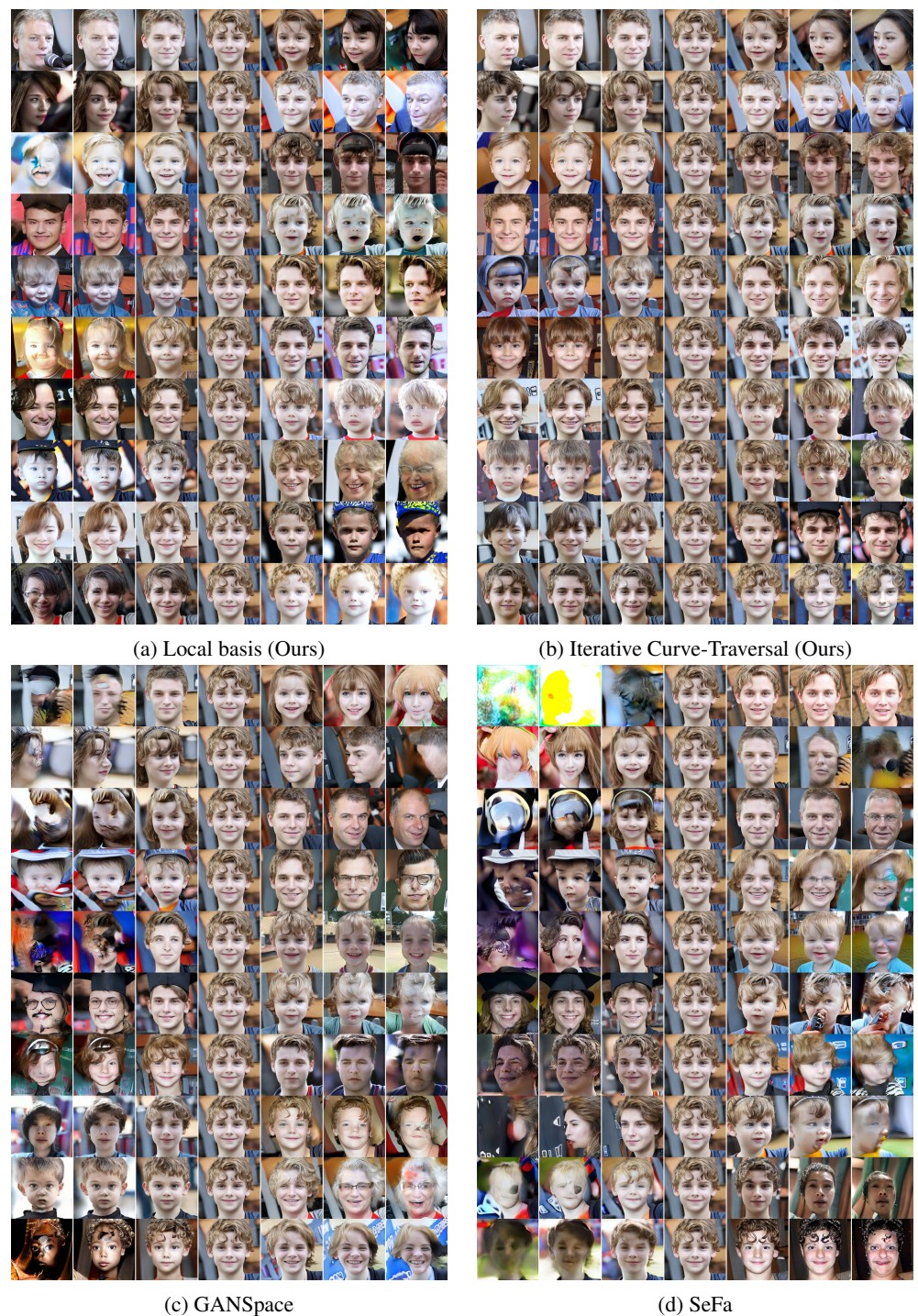

(a) Local basis (Ours)

(b) Iterative Curve-Traversal (Ours)

(c) GANSpace

(d) SeFa

Figure 12: **Additional Robustness Test results** of each Latent Traversal methods along the first 10 components. For each traversal methods, each row corresponds to a latent traversal of perturbation up to 12. Compared to the global methods, GANSpace and SeFa, even Local Basis with linear traversal (Fig 12a) shows more stable traversal on images. Moreover, Local Basis with Iterative Curve-Traversal (Fig 12b) rarely shows any collapse under the latent traversal of 12 along the curve.

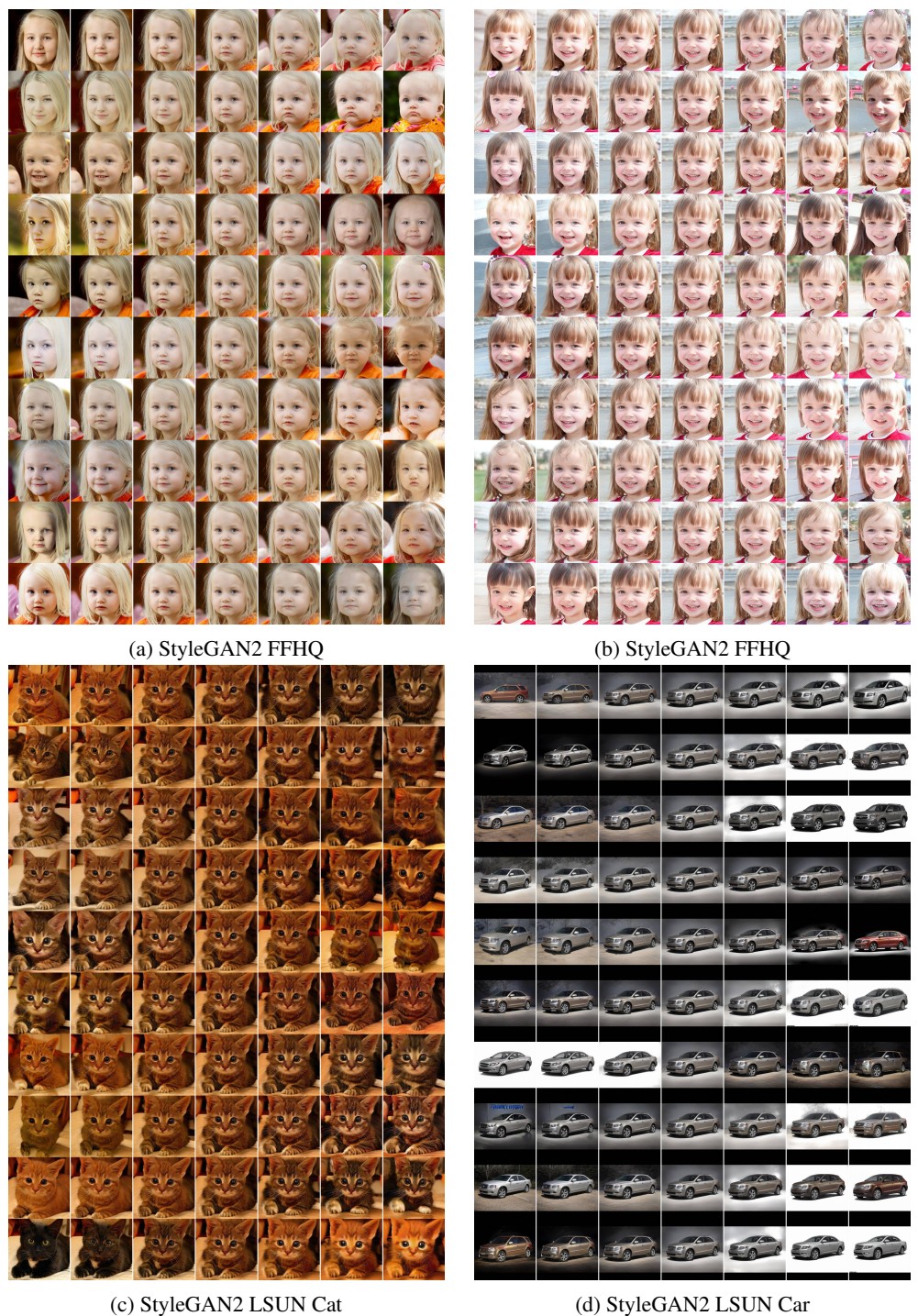

(a) StyleGAN2 FFHQ                    (b) StyleGAN2 FFHQ

(c) StyleGAN2 LSUN Cat               (d) StyleGAN2 LSUN Car

Figure 13: **Additional examples of Semantic Factorization without layer restriction**, i.e. Latent traversal along the first 10 components of Local Basis with a moderate perturbation of up to 5. Local Basis finds diverse and natural-looking semantic variations on each dataset.

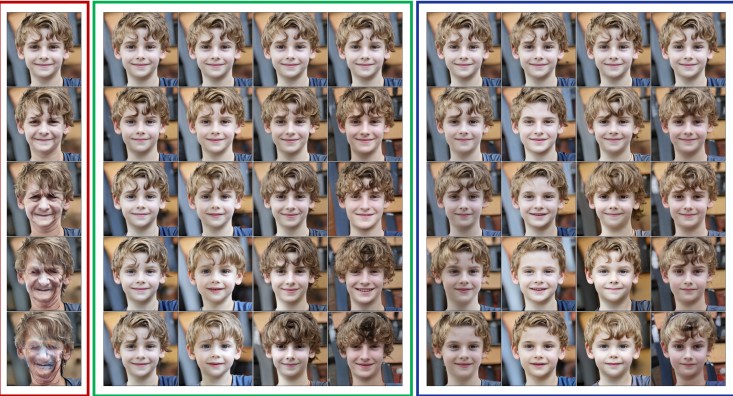

Figure 14: **Iterative Curve-Traversal guided by global basis only at departure** from StyleCLIP for the semantics of *old*. Contrary to Fig 7, Iterative Curve-Traversal follows the global basis only at departure. After that, the departure direction is chosen by the similarity to the previous departure direction. **Left**: Linear traversal along global basis. **Middle**: Iterative Curve-Traversal of fixed stepsize (Stepsize = (0.02, 0.04, 0.08, 0.16)). **Right**: Stochastic Iterative Curve-Traversal (Stepsize is sampled from Uniform Noise on [0.05, 0.15])

## H IMPLEMENTATION DETAILS FOR SEC 3.4

In Sec 3.4, we utilize the global basis from StyleCLIP (Patashnik et al., 2021) defined on $\mathcal{W}^+$, the layer-wise extension of $\mathcal{W}$ introduced in (Abdal et al., 2019; Patashnik et al., 2021). To be more specific, since the synthesis network in StyleGAN has 18 layers, we obtain an extended latent code $\mathbf{w}^+ \in \mathcal{W}^+$ defined by the concatenation of latent codes $\mathbf{w}_i \in \mathcal{W}$ of dimension 512 for each $i$-th layer and it can be described as follows:

$$\mathbf{w}^+ = (\mathbf{w}_1, \mathbf{w}_2, \cdots, \mathbf{w}_{18}) \in \mathbb{R}^{512 \times 18}. \tag{20}$$

Note that our Iterative Curve-Traversal originally defined on $\mathcal{W}$ has a canonical extension to $\mathcal{W}^+$ without additional changes in structures or methodologies.

For implementing the stochastic Iterative Curve-Traversal introduced in Sec 3.4, we firstly find a global basis on $\mathcal{W}^+$ using StyleCLIP (Patashnik et al., 2021) which implies a given semantic attribute in the form of text (e.g. old). Now denote the global basis by $\mathbf{v}_{global}^+$, which can be represented as follows:

$$\mathbf{v}_{global}^+ = (\mathbf{v}_1^{global}, \mathbf{v}_2^{global}, \cdots, \mathbf{v}_{18}^{global}). \tag{21}$$

Then we perform the (extended) Iterative Curve-Traversal following $\mathbf{v}_{global}^+$, equipped with a stochastic movement for each step. In practice, we consider two options to choose a traversal direction for each step; First, follow the direction most similar to the previously selected basis (as Algorithm 2), except for the first iteration. Note that we compute the similarity between the local basis and the global basis only at once when choosing the first traversal direction. Second, follow the direction most similar to the given global basis. This is slightly different from our Algorithm 2, however, we empirically verify that setting the exploration in that way leads to a more desirable image change.

Fig 14 shows that the first method still preserves the image quality well, but it does not guarantee that the desired direction of image change, namely 'old'. We speculate the reason why such phenomenon occurs is that most of the information contained in the meaningful global basis disappears after the first step (a unique, direct comparison to the global basis), although our methodology guarantees that the latent code does not escape from the manifold and achieve a high image quality. Nevertheless, Fig 15 shows that the second method for the stochastic Iterative Curve-Traversal can change a given facial image in a very high quality and various ways.

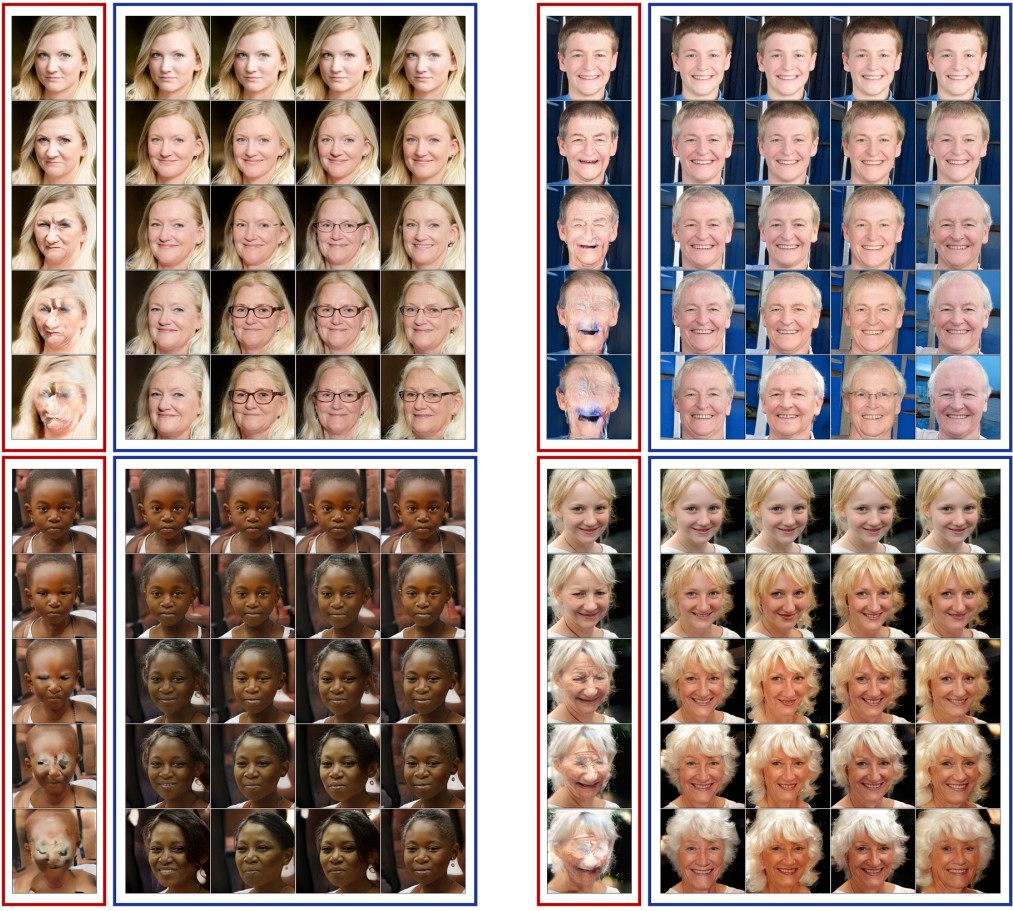

Figure 15: **Additional Examples of Stochastic Iterative Curve Traversal** guided by the global basis from StyleCLIP for the semantics of *Old*. **Left**: Linear traversal along global basis. **Right**: Stochastic Iterative Curve-Traversal

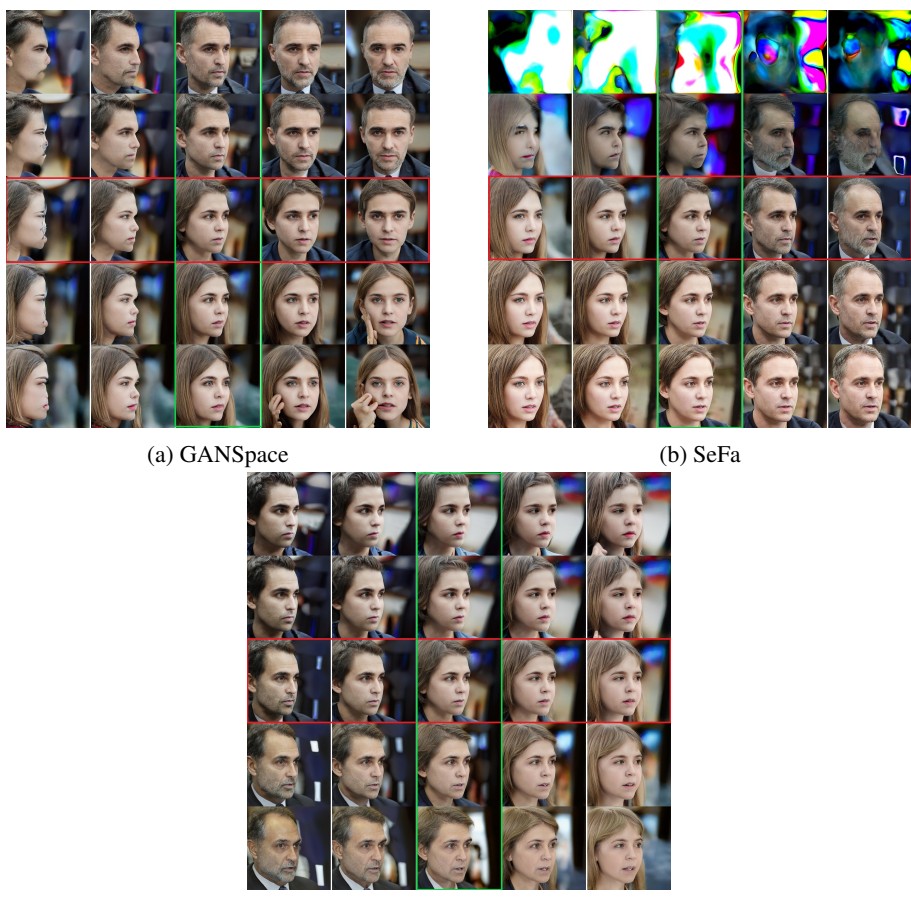

Figure 16: **Subspace traversal with two directions** on $\mathcal{W}$-space of the StyleGAN2. The horizontal (red box) and vertical (green box) axes correspond to the 1st and 2nd directions of each method.

## I   SUBSPACE TRAVERSAL

In Section 4, we proved that the $\mathcal{W}$-space in StyleGAN2 is warped globally. Specifically, the subspace of traversal direction generating principal variation in the image changes severely as we vary the starting latent variable $\mathbf{w}$. To verify the claim further, we visualize the subspace traversal on the latent space $\mathcal{W}$. The subspace traversal denotes a simultaneous traversal in multiple directions. In this paper, we visualize the two-dimensional traversal,

$$\textit{Subspace Traversal}^{\mathbf{w}}_{(i,j)}(x,y) = G\left(\mathbf{w} + \frac{x}{N}\mathbf{v}^{\mathbf{w}}_i + \frac{y}{N}\mathbf{v}^{\mathbf{w}}_j\right) \tag{22}$$

where $\mathbf{w} = f(\mathbf{z})$ and $G$ denotes a subnetwork of the given GAN model from $\mathcal{W}$ to the images space $\mathcal{X}$. Since the disentanglement into the linear subspace implies the commutativity of transformation (Pfau et al., 2020), the subspace traversal can be a more challenging version of linear traversal experiments.

Fig 16 and Fig 17 show results of the subspace traversal for the global basis and Local Basis. Starting from the center, the horizontal and vertical traversals correspond to the 1st and 2nd directions of each method. The same perturbation intensity per step is applied for both directions. When restricted to the linear traversal (red and green box), the GANSpace shows relatively stable traversals. However, the traversal image deteriorates at the corner of the subspace traversal. By contrast, Local Basis shows a stable variation during the entire subspace traversal. This result proves that the global basis is not well-aligned with the local-geometry of the $\mathcal{W}$ manifold.

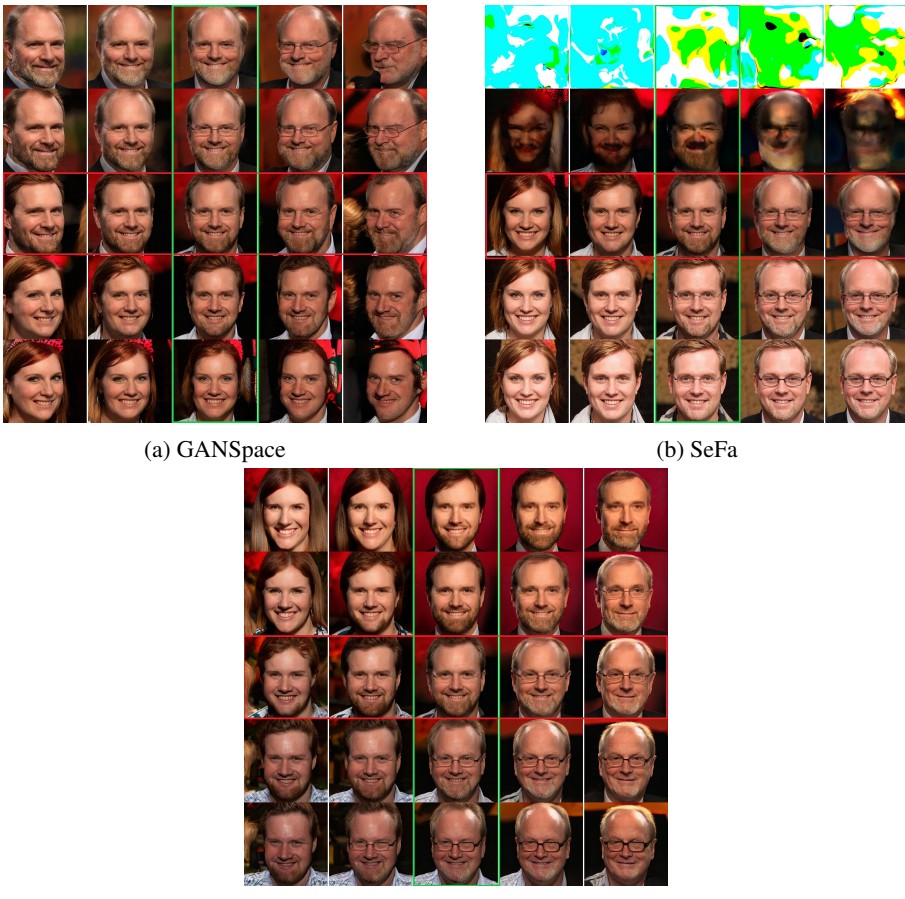

(a) GANSpace

(b) SeFa

(c) Local Basis (Ours)

Figure 17: **Subspace traversal with two directions** on $\mathcal{W}$-space of the StyleGAN2. The horizontal (red box) and vertical (green box) axes correspond to the 1st and 2nd directions of each method.

## J    LOCAL BASIS ON OTHER MODELS

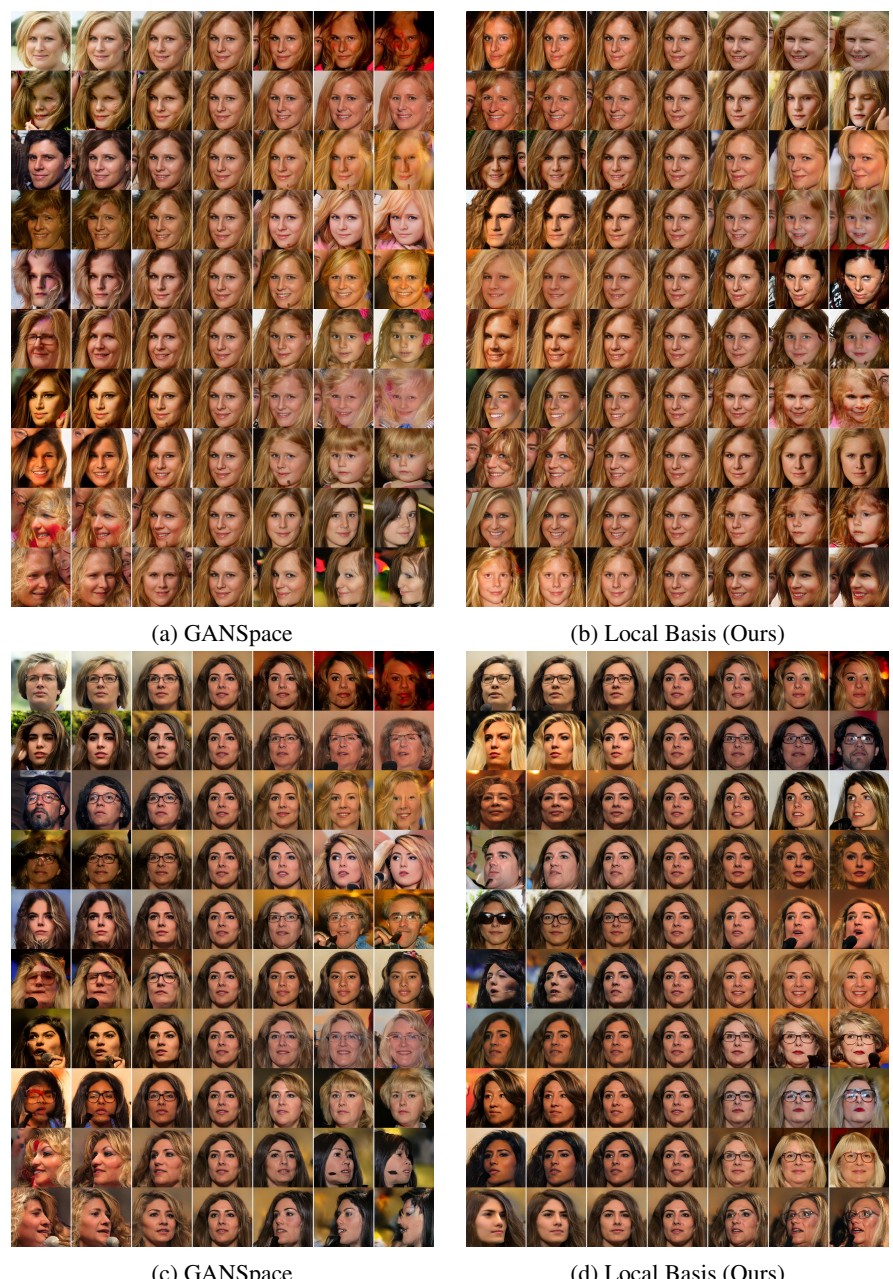

(a) GANSpace

(b) Local Basis (Ours)

(c) GANSpace

(d) Local Basis (Ours)

Figure 18: **Comparison of GANSpace and Local Basis on StyleGAN-FFHQ (Karras et al., 2019)** Each traversal image is generated along the first 10 components of each method with a perturbation of up to 5.

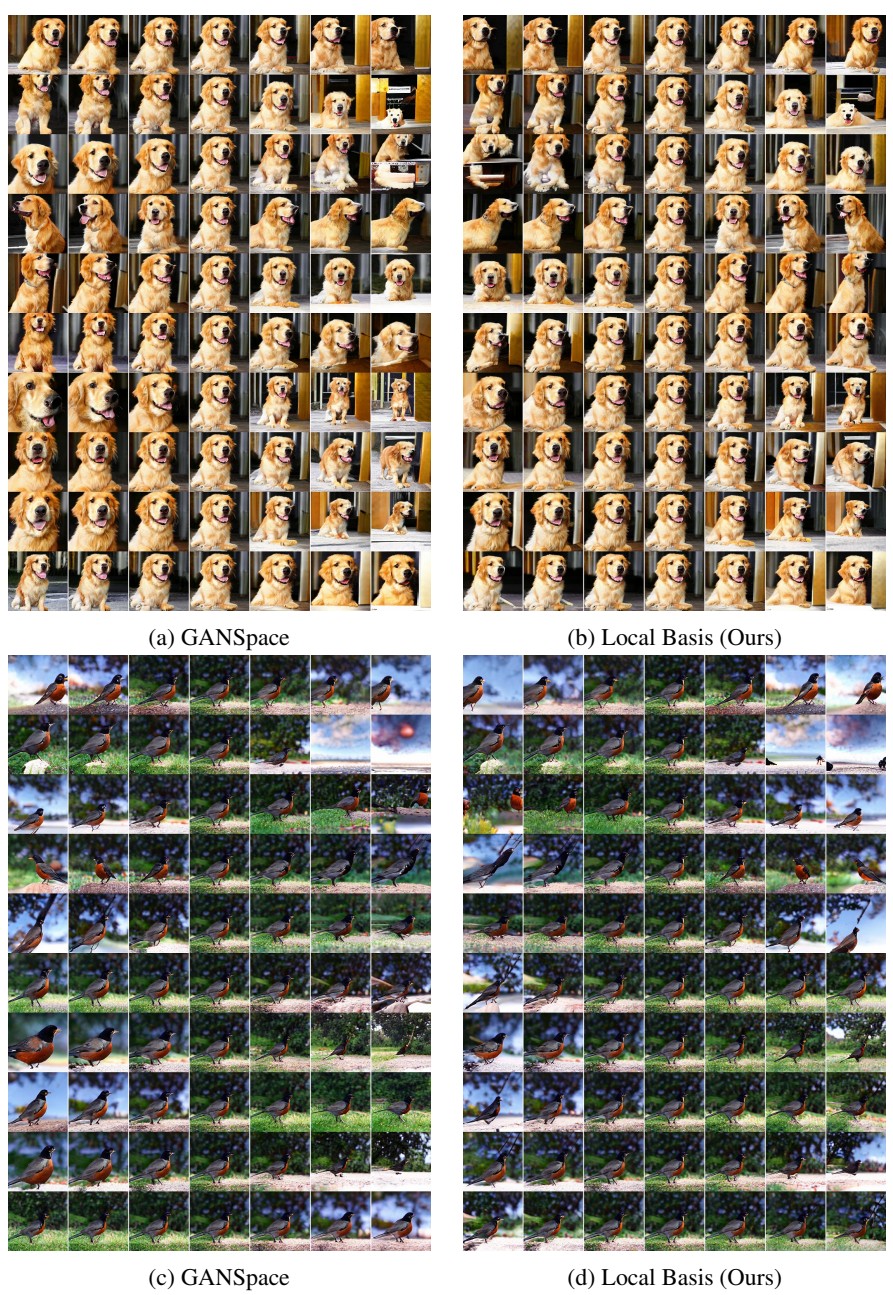

(a) GANSpace

(b) Local Basis (Ours)

(c) GANSpace

(d) Local Basis (Ours)

Figure 19: **Comparison of GANSpace and Local Basis on BigGAN-512 (Brock et al., 2018)** Each traversal image is generated along the first 10 components of each method with a perturbation of up to 3.

