# OpenReview forum: "Do Not Escape From the Manifold: Discovering the Local Coordinates on the Latent Space of GANs"
_ICLR.cc/2022/Conference — ICLR 2022 Poster_

### Official Review · Reviewer_sKqG · 2021-10-21

**Correctness:** 3
**Technical Novelty And Significance:** 3
**Empirical Novelty And Significance:** Not applicable
**Recommendation:** 8
**Confidence:** 3

**Main Review:**

Overall, they did well in this work and are high in potential.

STRENGTHS:
- End-to-end, the paper stays focused on a consistent theme via a well-organized summary of their work.
- Visualizations show concepts well
- There is a nice distribution of research paper components (e.g., figures, tables, proper intro, background, conclusion, and such).
- The idea is interesting and relevant.

WEAKNESSES:
- Figures have text that is difficult to read (text in figures should match the font that is used in the paper's body, in style and size).
- Equations look okay, but there are several components that should not be variable (i.e., italicized). For instance, "Local Basis (...)..." in Eq 4 should be wrapped in \text{Local Basis}-- there are many instances like this. Make sure only variables are italicized in equations.
- Fig 1, 4, 6, 8 are low in quality. Increase the resolution and be sure to use vectorized images (e.g., PDF).
- They do not reference Figure 1 in the text. Also, the first figure referenced in the text is in the supplemental material. Referring to supplemental to further support claims is okay, however, it seems like Fig 9 should be in the major paper (i.e., or they should change the paragraph referring to it not to depend on the supplemental)
- Figures are not well explained
   - the conditions for which they interpolated other methods are unclear-- why some cases there are clear variables being swept (e.g., pose of the face), but in others, it is not quite clear. Why does this vary from method to method to the extent it does? Sure, the proposed might work well at staying within photorealistic ranges but is the reason that the other methods do not be that sweeping used was not optimal for each other SOA method
--

**Summary Of The Paper:**

The authors proposed Local Basis: a purposeful traversal direction based on the local-geometry of the intermediate latent space of a GAN. To support the theory of Local Basis, the global-geometry of the latent space, along with an iterative traversal method to trace the latent space, moves apart from the space in a piece-wise, step-by-step manner. Their results show that Local Basis decomposes an image's semantics, providing more stability in transforming images via the proposed iterative traversal. Following the suggested evaluation of the W-space of StyleGAN2 reveals that the W-space is distorted at the global level: the global method, thus, limits global consistency while in W-space.


**Summary Of The Review:**

The paper is a marvelous piece: well put together, professionally presented, relevant and interesting topic, excessive experiments and analysis, and, mostly, well written.

DISCLAIMER: the mathematics in this paper I am not too familiar with-- there appeared to be consistent use in variables, each properly defined, and expressed in the body of text coherently. If another review can validate the math, it would be great (i.e., my review does not include deeply validating the math, for low-rank I have minimal experience with.

My primary concern is that the analysis is limited (not in figures, but in an explanation). Also, the story could be told better. Note that the two aforementioned concerns go hand in hand: with a better description of the image montages (i.e., what to look for, highlight the pros, also show some failure modes). Share thoughts on why results are better; same with failure modes (which are not currently highlighted).

Thus, the figures are sufficient, but the intuitive insight by breaking the figures and plots down is severally lacking (both in-text body, caption, and in the visual themselves). This would likely yield a better story, end-to-end, for it might bring the motivation full circle: as is, I am not seeing it go full circle, as all the best stories do.

With that, assuming the math is correct (per another reviewer validating), the pros slightly outweigh the cons, so I am neutral (or slightly above, since there is no middle chose :)

---

> ### Author Response · Authors · 2021-11-18
> **Thank you for questions and feedback**
>
> We thank the reviewer for the positive feedback. We are delighted that the reviewer considers our work is “high in potential”. Moreover, we appreciate the detailed advice on the visualization, organization, and presentation of our work. \
> Below we address specific questions and comments:
>
> __A. Visualization and presentation__
>
> Thank you for the good advice. In our updated version, we corrected all the deficiencies of figures and equations. To be more specific, we improved the quality and readability of images by increasing the resolution and font size. We revised the non-variable instances which were italicized in equations. In addition, each graph in Fig 4, 8 is revised to have a different marker and line style so that it could be easily distinguished. The details are as follows:
>
> > Figure 1(a), 2, 8 - For readability, the size of the legend is increased.\
> > Figure 1(b), 5, 6 - The text font is changed to the same font as the body, and the font size and thickness are increased. And the overall design of Fig 6 is improved\
> > Figure 4 - Each method is marked with a different marker so that it could be easily distinguished, and the font size is increased.\
> > Eq 4 - _Local Basis_ $\rightarrow$ Local Basis\
> > Eq 7 - _Traversal_ $\rightarrow$ Traversal
>
> __B. Organization__
>
> We agree with the reviewer that Fig 9 would be better in the major paper. Regrettably, due to space limitations in the main paper, we changed the paragraph to not depend on the supplemental. In addition, we added the reference to Fig 1 in the major paper.
> > Updated at line 4 in Page 4
>
> __C. The unclarity of the conditions for which we interpolated other methods__
>
> For each image traversal result (Fig 3, 5, 6, 7), each sample is generated while fixing the perturbation intensity regardless of the traversal methods and directions. For Stochastic Iterative Curve-Traversal in Fig 7, we matched the expectation of the perturbation intensity. Here, the perturbation intensity is defined as the traversal path length in $\mathcal{W}$ (Sec 3.3). To be more specific, given a pretrained GAN model $M : \mathcal{Z} \xrightarrow{f} \tilde{\mathcal{W}} \xrightarrow{G} \mathcal{X}$, the linear traversed image $\mathbf{x}_{\mathbf{w}, \mathbf{v}}^{I}$ from $\mathbf{w}$ along $\mathbf{v}$ with perturbation intensity $I$ is obtained as follows:
>
> $$
> \mathbf{x}_{\mathbf{w}, \mathbf{v}}^{I} = G(\mathbf{w} + I \cdot \mathbf{v})
> $$
>
> where the traversal direction $\mathbf{v}$ is a normalized unit vector.
>
> We believe that comparing latent traversal methods while matching the perturbation intensity $I$ is a fair comparison because they are applied to the same latent space $\mathcal{W}$. In particular, we applied the $I = 12$ for Fig 3, $I = 10$ for Fig 5a, $I = 5$ for Fig 5b, $I = 8$ for Fig 5c, $I = 10$ for Fig 5d, and $I = 4$ for Fig 7.
>
>
> __D. Additional explanation of figures__
>
> Thank you for the good advice. To make a better description and comparison of the qualitative traversal results, we added an additional explanation of Fig 5 in the Semantic Factorization paragraph of Sec 3.3.
> > Updated at Section 3.3,  __Semantic Factorization__ paragraph "In particular, ... any of those problems."

---

> > ### Comment · Reviewer_sKqG · 2021-11-23
> > **Rebuttal response**
> >
> > All-around, nice work!

---

### Official Review · Reviewer_Pg5m · 2021-10-27

**Correctness:** 4
**Technical Novelty And Significance:** 2
**Empirical Novelty And Significance:** 3
**Recommendation:** 6
**Confidence:** 4

**Main Review:**

Disentanglement of a representation often stems from intuitions gathered from linear models, namely that we should expect that different aspects of data may change as we move along a chosen Euclidean coordinate axis (similar to principal components). This paper challenges this view and argue that we should only locally follow a specific direction, and that this direction should come from the subspace spanned by the top singular vectors of the Jacobian of the generator (the specific paper investigate GANs, but the idea is general). It is trivially seen (Proposition 1) that if we consider samples from a 'small' Gaussian in latent space, then looking at the directions of the top singular vectors of the Jacobian amounts to local PCA in observation space (follows from Taylor's theorem).

This analysis is sound and reasonable. It is, however, not particularly novel. In the context of VAEs (rather than GANs), there is a large body of work that equip the latent space with the Riemannian pull-back metric associated with the generator, i.e. $G(z) = J(z)^T J(z)$, where $J$ is the Jacobian, and $G$ is then the metric tensor. Existing work then study the general identifiability problem associated with latent variable models; this is arguably a more general problem than the disentanglement problem. Riemannian quantities such as geodesics then serve similar purposes to the traversels proposed in the present paper, though geodesics are perhaps better understood. I speculate if indeed the proposed traversel method isn't just a first-order Euler method for solving the initial value problem associated with geodesics.

Related work in this regard (these groups of authors has published significantly more on the topic, but I just list the most commonly cited papers):
1. Latent Space Oddity: on the Curvature of Deep Generative Models. G. Arvanitidis, LK. Hansen & S. Hauberg. ICLR 2018. This was the first paper to study the Riemannian geometry of VAEs; a key technical difference to the paper at hand is that Arvanitidis et al. considered stochastic generators, while the current paper considers deterministic generators.
2. N. Chen, A. Klushyn, R. Kurle, X. Jiang, J. Bayer & P. Smagt. Metrics for deep generative models. AISTATS 2018. Similar to the ICLR paper above except it considers deterministic generators.
3. H. Shao, A. Kumar & PT. Fletcher. The Riemannian geometry of deep generative models. CVPR Workshops 2018. Similar to paper 2.

These papers focus on the identifiability problem and not the disentanglement problem, so in that regard the present paper has a novel contribution. This, however, seems rather minor as identifiability is arguable a more general and harder problem.

I have already mentioned that the traversal algorithm seems quite similar to geodesic computations, e.g. akin to the algorithm proposed in paper 3 above. It also seems very similar to the Brownian motion / diffusion sampling proposed in this paper:

4. Variational Autoencoders with Riemannian Brownian Motion Priors. D. Kalatzis, D. Eklund, G. Arvanitidis & S. Hauberg. ICML 2020

The main difference seems to be that paper 4 solves and SDE (they add a bit of Gaussian noise in each step), while the present seems more like an ODE solver.

The current paper, like so many other disentanglement papers, has a somewhat ad hoc empirical evaluation strategy. This is fair given the complexity of the topic. Results seems encouraging, but it's really difficult to get a feel for which approaches are better. I don't think this is the authors fault, but it still seems a bit disappointing.

I should also note that I did not really understand the point of the warpage evaluation. It isn't clear to me what the authors aim to achieve in this part of the paper.

All that being said, I think the paper is nicely presented and the empirical results are generally quite nice (unlike the above papers that are more proof-of-concepts).

*Post-rebuttal*
I think the authors answered my key concerns during our discussions, and I trust that the paper will be adjusted accordingly. I have therefore bumped my score up a bit to recommend acceptance.

**Summary Of The Paper:**

The paper propose to explore disentanglement locally through the top singular vectors of the Jacobian of the generator of a GAN. This contrast the popular view that disentanglement should follow Euclidean coordinate axes in the latent space. An algorithm for traversing the latent space is proposed.

**Summary Of The Review:**

The key ideas behind the paper has appeared in a series of papers elsewhere, and these arguably provide a deeper understanding of the problem than what is discussed here. That makes it hard to recommend acceptance. That being said, the paper is well-written and the empirical results are interesting, hence the borderline/reject score.

---

> ### Author Response · Authors · 2021-11-18
> **Thank you for questions and feedback**
>
> __D. Additional explanation of the warpage evaluation.__
>
> We intended to present the aim of Sec 4 (Warpage section) from the top of Sec 4 to “Grassmannian Manifold” paragraph. We are sorry it was not presented clearly. The additional explanations for the main intuition of Sec 4 are as follows (We revised the submitted paper accordingly.):
> 1.	From Sec 3, we showed that Local Basis corresponds to the generative factors of data. Hence, the linear subspace spanned by Local Basis describes the local principal variation of the image.
> 2.	The linear subspace spanned by Local Basis is the same as the tangent space of the approximating submanifold of $\mathcal{W}$ in Eq 5. Therefore, we assess the global disentanglement property by assessing the consistency of the tangent space at each $\mathbf{w} \in \mathcal{W}$. We refer to the inconsistency of the tangent space as the warpage of the latent manifold.
> 3.	The consistency of the tangent space is assessed by measuring the Grassmannian distance between two tangent spaces. As a reminder, for a vector space $V$, the Grassmannian manifold $\text{Gr}(k, V)$ is defined as the set of all $k$-dimensional linear subspaces of $V$.
> > Updated at the top of Section 4
>
>
> __References__
>
> [1] Latent Space Oddity: on the Curvature of Deep Generative Models, G. Arvanitidis, LK. Hansen & S. Hauberg. ICLR 2018.\
> [2] Metrics for deep generative models, N. Chen, A. Klushyn, R. Kurle, X. Jiang, J. Bayer & P. Smagt. AISTATS 2018. \
> [3] The Riemannian geometry of deep generative models, H. Shao, A. Kumar & PT. Fletcher. CVPR Workshops 2018. \
> [4] Variational Autoencoders with Riemannian Brownian Motion Priors, D. Kalatzis, D. Eklund, G. Arvanitidis & S. Hauberg. ICML 2020. \
> [5] StyleSpace Analysis: Disentangled Controls for StyleGAN Image Generation, Z. Wu, D. Lischinski & E. Shechtma. CVPR 2021. \
> [6] Challenging common assumptions in the unsupervised learning of disentangled representations, Locatello, Francesco, et al. ICML 2019. \
> [7] Ganspace: Discovering interpretable gan controls, E. Härkönen, A. Hertzmann, J. Lehtinen & S. Paris. NeurIPS 2020.

---

> ### Author Response · Authors · 2021-11-18
> **Thank you for questions and feedback**
>
>
> __B. Traversal algorithm__
>
> - Geodesic Shooting in [3]
>
> The main difference between Geodesic Shooting in [3] and our Iterative Curve-Traversal lies in how they choose the next traversal directions. As mentioned in A, our work suggests the usefulness of _Local Basis_, the set of orthonormal vectors corresponding to the local coordinate grids. Therefore, our Iterative Curve-Traversal follows Local Basis at each $\mathbf{w}$-iterates. On the other hand, Geodesic Shooting does not have any preferable basis of the tangent space. Hence, Geodesic Shooting projects the previous traversal directions on the tangent space of iterates. In this regard, we consider our Iterative Curve-Traversal is different from Geodesic Shooting. As for the interpretation of the Euler method, we agree that our method reminds of the iterative one-dimensional variant of the Euler method while the one-dimension, where the Euler method takes a forward step, is chosen for each iteration as described in Eq 12.
>
> - Brownian motion / diffusion sampling proposed in [4]
>
> Meanwhile, [4] argues that it is better to regard the latent space of VAE as a Riemannian manifold instead of the Euclidean space and they use a Riemannian Brownian motion as an alternative prior for the latent variable.
> Moreover, their sampling procedure requires following a stochastic path on the manifold and it enables the latent vectors to be sampled from the high density region.
> These concepts are quite similar to our paper, in terms of that the latent space can be not flat and the latent vector should be sampled from the manifold.
>
> Nevertheless, there are significant differences between [4] and ours.
> While [4] discuss how to 'sample' the latent vectors from a non-trivial prior, our work does not deal with a prior for the intermediate latent space $\mathcal{W}$.
> Also, the metric tensor $J_{a}$ in [4] forces the latent vector to move along the high density region of the manifold, while the Jacobian matrix of the mapping network $f$ forces the latent vector to move along the range of $f$ ($\mathcal{W}$-manifold).
> In our opinion, the difference in the architectures of VAE and StyleGAN literature is the main reason why these two approaches, [4] and ours, can not be considered as the same.
> While VAE has an explicit prior for the latent variable, StyleGAN samples $z$ from the standard Gaussian prior and obtains $w$ by mapping $z$ through $f$.
> Since the mapping network $f$ consists of the 8-layer neural networks, there is no closed-form representation for the distribution of $f(z) = w \in \mathcal{W}$ and it is hard to recognize the high density region in the $\mathcal{W}$-manifold.
> In this regard, our _Local Basis_ can be useful when the probabilistic density is unknown or intractable, e.g. the intermediate latent space $\mathcal{W}$ in StyleGAN.
>
>
>
> __C. Empirical evaluation strategy__
>
> We agree that it would have been better to evaluate the semantic factorization property of Local Basis quantitatively. To make a fairer comparison out of the qualitative evaluation of unsupervised disentanglement algorithms, we utilized the samples released by the authors of [7] in Fig 5. The authors of [7] provide the pairs of samples, perturbation directions, and the corresponding attributes. We compared the semantic factorization of GANSpace and Local Basis of the highest cosine similarity. In Fig 5, each column corresponds to the same intensity of perturbation. Moreover, we attached the additional traversal examples along the first ten components of Local Basis to the supplementary material.

---

> > ### Comment · Reviewer_Pg5m · 2021-11-20
> > **Thanks for the clarifications**
> >
> > I appreciate the replies.

---

> ### Author Response · Authors · 2021-11-18
> **Thank you for questions and feedback**
>
> We thank the reviewer for the thoughtful feedback. We think that the reviewer raised several excellent questions and the suggested references would be helpful to our further research.
>
> Below we address specific questions and comments:
>
> __A. Previous works about latent space equipped with the Riemannian pull-back metric.__
>
> The summary of the suggested references are as follows:
>
> > [1] extends the Riemannian metric on the latent space to the stochastic generator. The induced geodesic trajectory is computed by solving the ODEs with the numerical solver.  \
> [2] deals with how to create a geodesic connecting two points in the latent space given the pull-back metric. Notably, the geodesic trajectory is parametrized with a neural network. \
> [3] suggests the geodesic interpolation, parallel translation, and geodesic shooting on the generated image manifold. In particular, the geodesic trajectory is discovered by performing gradient descent of $\mathbf{z}$-iterates on the arc-length energy functional.
>
> These works and ours have similar mathematical forms because of the common use of differential, the induced linear map between tangent spaces. Nevertheless, we believe they differ in their motivations and goals. In our opinion, the idea of assigning the pull-back metric on the latent space is motivated by the assumption that the metric on the latent space should reflect the degree of changes in the generated image. In this regard, [1, 2, 3] evaluate the smoothness of geodesic interpolation on the image manifold. Also, [3] demonstrates that the $k$-means clustering algorithm with pull-back metric provides better clustering.
>
> On the other hand, our work utilizes the differential to estimate the intermediate latent space $\mathcal{W}$. In particular, our work shows that the local principal components (_Local Basis_) of $\mathcal{W}$ corresponds to the robust and semantic-factorizing traversal directions. It is worth noting that _Local Basis_ do not utilize the synthesis network $G : \mathcal{\tilde{W}} \rightarrow \mathcal{X}$, which is directly connected to image synthesis.
> Moreover, this estimation of intermediate latent space enables an unsupervised quantitative analysis of the global disentanglement property through the Grassmannian metric in Sec 4. In Sec 4, we showed that the limited success of existing global methods is due to the global warpage of $\mathcal{W}$. In this respect, it is still challenging to find the best latent space in GAN for manipulating a generated image. In the cases of StyleGAN and StyleGAN2, various latent spaces such as $\mathcal{W}$, $\mathcal{W^{+}}$, and $\mathcal{S}$ [5] were proposed for the better disentanglement property. The proposed evaluation scheme may provide the criteria for the global disentanglement property of the arbitrary latent space.
>
>
> Unfortunately, we could not fully understand the relationship between these papers and the identifiability problem. To the best of our knowledge, we understand the identifiability problem as the one discussed in [6]. We are sorry to bother you, but we would appreciate it if you could explain in more detail.

---

> > ### Comment · Reviewer_Pg5m · 2021-11-20
> > **Identifiability**
> >
> > Thanks for the follow-up. I agree with the analysis that you provide above.
> >
> > I agree that my comments regarding identifiability were not entirely clear and I apologize for that. I will try to be a bit more clear this time. In general in latent variable models, the latent variables are not identifiable, i.e. there are many different configurations of the latent space that can give rise to the same distribution. This is the mathematical reason why different training runs can yield different (but equally good in terms of likelihood) latent configurations. This is the key reason why disentanglement is not possible in terms of Euclidean axes (this discussed in various places, e.g. the papers by Locatello et al.). The papers I referred to above call on differential geometry to provide a solution: since geodesic distances are defined through the metric of the ambient observation space, they are invariant to reparametrizations of the latent space, and therefore geodesic distances are identifiable. In terms of disentanglement, this imply that as long as we disentangle using the underlying Riemannian metric, then results are invariant to reparemetrizations of the latent space, and the disentangling factors are identifiable.
> >
> > As far as I can see, your approach amounts to walking along local principal directions according to the metric of the observation space, which is a form of local principal geodesic analysis according to the same Riemannian metric as in papers 1-3 above.

---

> > > ### Author Response · Authors · 2021-11-23
> > > **Thanks you for the additional explanations.**
> > >
> > > Thank you for the additional explanations and for suggesting an interesting perspective.
> > >
> > > We consider the geodesic perspective provides a fair additional interpretation of our work. Because the metric of the observation space $\mathcal{X}$ is given from the ambient space, it is identifiable to find a geodesic curve on the latent space that corresponds to the principal tangent vectors of the observation space.
> > > Specifically, the identifiability is defined in terms of the image of differential $\mathbf{v_x} = df_\mathbf{z} \mathbf{(u_z) \in T_{x} \mathcal{X}}$.
> > >
> > > Here, the observation space $\mathcal{X}$ would be the image space for [1-3] and the intermediate latent space $\mathcal{W}$ for our work. Therefore, the geodesic perspective provides an interpretation of our Local Basis as the identifiable method for finding the principal tangent vectors of the intermediate latent space $\mathcal{W}$.
> > >
> > > However, in our opinion, the geodesic perspective still requires the correspondence between the principal tangent vectors of the observation space and the generative factors of data for the identifiability discussed in [6].
> > > Formally, the generative factors of data are defined as the axes of the factorized prior distribution $p(\mathbf{z})$. The correspondence is demonstrated empirically for the intermediate latent space $\mathcal{W}$ in our work and for the image space in [8] through the semantic factorization property. In [1-3], the Riemannian metric is analyzed as the whole, not as the decomposed form for the disentanglement. We believe the theoretical approach for the correspondence between the principal tangent vectors and the generative factors
> > > would be an interesting further research.
> > >
> > > __References__ \
> > > [1] Latent Space Oddity: on the Curvature of Deep Generative Models, G. Arvanitidis, LK. Hansen \& S. Hauberg. ICLR 2018. \
> > > [2] Metrics for deep generative models, N. Chen, A. Klushyn, R. Kurle, X. Jiang, J. Bayer \& P. Smagt. AISTATS 2018. \
> > > [3] The Riemannian geometry of deep generative models, H. Shao, A. Kumar \& PT. Fletcher. CVPR Workshops 2018. \
> > > [6] Challenging common assumptions in the unsupervised learning of disentangled representations, Locatello, Francesco, et al. ICML 2019. \
> > > [8] A spectral regularizer for unsupervised disentanglement, A Ramesh, Y Choi \& Y LeCun. ICML 2019.

---

> > > > ### Comment · Reviewer_Pg5m · 2021-11-23
> > > > **Thanks for the follow-up**
> > > >
> > > > I agree with the analysis presented here and will take it into consideration during the discussions among reviewers.

---

### Official Review · Reviewer_v9kk · 2021-11-04

**Correctness:** 4
**Technical Novelty And Significance:** 3
**Empirical Novelty And Significance:** 3
**Recommendation:** 6
**Confidence:** 5

**Main Review:**

The idea is simple and convincing. The key is to use the principal subspace of the Jacobian to extract  attribute representations. The novelty lies in two aspects: 1) the authors study the Jacobian of the mapping network, as opposed to previous works and 2) the editing path is iteratively formed by taking the most similar direction  of adjacent tangent spaces (principal subspaces of the Jacobians).

A very relevant NeurIPS work is missed.

Low-Rank Subspaces in GANs
https://arxiv.org/abs/2106.04488

In this paper, the authors reviewed the related work using the Jacobians. Please cite those related papers. In this work, the authors show that the principal subspace of the Jacobian may fail to capture disentangled attribute directions and result in artifacts.  They use the null space of the Jacobian with respect to complementary region  to improve the performance. Please discuss this problem in your paper.

**Summary Of The Paper:**

The authors proposed a geometric method for image editing with pretrained StyleGAN2. The idea is simple. Editing is performed by constraining the editing path on the manifold. The simplest way of making this is to use the tangent space. A natural realization of the tangent space is the principal subspace of the Jacobian. The authors harness the Jacobian regarding the mapping from $z$ to $w$. The extensive experimental results verify the effectiveness of the method.

**Summary Of The Review:**

A simple and effective algorithm supported by geometry. But the author did not seriously discuss the drawback revealed by the relevant paper. This part should be improved.

---

> ### Author Response · Authors · 2021-11-18
> **Thank you for questions and feedback**
>
> We thank the reviewer for the positive feedback and for suggesting interesting related works. \
> [1] suggests a latent perturbation vector that can change only a particular area of the image. The idea is intuitive and convincing. [1] finds the principal vector of the Jacobian matrix from the input latent $\mathbf{z}$ to the target area. Then, the principal vector is projected into the null space of the Jacobian matrix from the input latent to the outside of the target area. The projection ensures that the latent perturbation does not cause changes outside the target area. \
> As the reviewer pointed out, [1] is very relevant to our work, so we added the comparison of [1] and ours in the related work section of the updated version. [1] reported that the principal subspace of the Jacobian may result in artifacts such as white spots. However, we could not find such a problem with Local Basis as you can see from the traversal examples in the main text and appendix. From the observation that GANSpace and SeFa do not show such a problem as well, we consider this to be an advantage of performing traversal in the intermediate latent space. Nevertheless, the isolated manipulation of [1] is impressive and would be helpful to our further research. Also, [2], [3], which are methods using Jacobians cited in [1], are added to the related work section.
> > Updated at Section 2,  __Jacobian Decomposition__ paragraph
>
> __Reference__
>
> [1] Low-Rank Subspaces in GANs, J. Zhu, R. Feng, Y. Shen, D. Zhao, Z. Zha, J. Zhou & Q. Chen. NeurIPS 2021. \
> [2] Human-in-the-loop differential subspace search in high-dimensional latent space, CH Chiu, Y Koyama, YC Lai, T Igarashi & Y Yue. ACM Transactions on Graphics (TOG), 39(4), 85-1. \
> [3] The Geometry of Deep Generative Image Models and its Applications, B Wang, CR Ponce. ICLR 2021.

---

> > ### Comment · Reviewer_v9kk · 2021-11-24
> > **Jacobian Decomposition**
> >
> > I read the updated part in the related work. I agree with you that it is  "an advantage of performing traversal in the intermediate latent space." I think the artifacts are avoided due to the Jacobians solved from the mapping network instead of the synthesis network in StyleGAN. This might be the key of using the principal subspace of the Jacobian correctly. If performed via the Jacobian from the synthesis network, it will incur the problem revealed in [1]. But this has little connection with GANSpace and SeFa.

---

> > > ### Author Response · Authors · 2021-11-26
> > > **Thank you for the additional comments.**
> > >
> > > Thanks for the additional comments.
> > >
> > > We agree with the reviewer that the global degradation of GANSpace and SeFa in Fig 3 might be irrelevant to the artifacts. Comparing Local Basis with [1], the reason for avoiding the artifacts might be the Jacobians from the mapping network, as the reviewer suggested. Meanwhile, we consider the Low-rank approximation of the Jacobian by PCP(Principal Component Pursuit) proposed [1] is an interesting approach. In our work, we verified the manifold hypothesis by observing the empirical distribution of singular vales of Jacobian in Fig 9. The sparsity of singular values supported the manifold hypothesis. On the other hand, the PCP approach can provide an estimate of the dimension of the approximating submanifold in Eq 5. Combining the PCP approach to our Local Basis might provide an interesting result for the intermediate latent manifold estimation.

---

### Decision · Program_Chairs · 2022-01-20

**Decision:**

Accept (Poster)

**Comment:**

The initial reviews for this paper were diverging. After the rebuttal all reviewers have reached the consensus of recommending the paper's acceptance. Some reviewers have concerns regarding the novelty of the paper, however they appreciate that the paper is ell written and the empirical results are interesting. Following the reviewers recommendation, the meta reviewer recommends acceptance. In the final version of the paper the authors are encouraged to strengthen the weaknesses discussion as requested by one of the reviewers.